# Semi-Supervised Vision-Centric 3D Occupancy World Model for Autonomous Driving

**Xiang Li, Pengfei Li, Yupeng Zheng, Wei Sun, Yan Wang**[*]**, Yilun Chen**[*]
Institute for AI Industry Research (AIR), Tsinghua University
`l-x21@mails.tsinghua.edu.cn; wangyan@air.tsinghua.edu.cn`

## Abstract

Understanding world dynamics is crucial for planning in autonomous driving. Recent methods attempt to achieve this by learning a 3D occupancy world model that forecasts future surrounding scenes based on current observation. However, 3D occupancy labels are still required to produce promising results. Considering the high annotation cost for 3D outdoor scenes, we propose a semi-supervised vision-centric 3D occupancy world model, **PreWorld**, to leverage the potential of 2D labels through a novel two-stage training paradigm: the self-supervised pre-training stage and the fully-supervised fine-tuning stage. Specifically, during the pre-training stage, we utilize an attribute projection head to generate different attribute fields of a scene (e.g., RGB, density, semantic), thus enabling temporal supervision from 2D labels via volume rendering techniques. Furthermore, we introduce a simple yet effective state-conditioned forecasting module to recursively forecast future occupancy and ego trajectory in a direct manner. Extensive experiments on the nuScenes dataset validate the effectiveness and scalability of our method, and demonstrate that PreWorld achieves competitive performance across 3D occupancy prediction, 4D occupancy forecasting and motion planning tasks.[1]

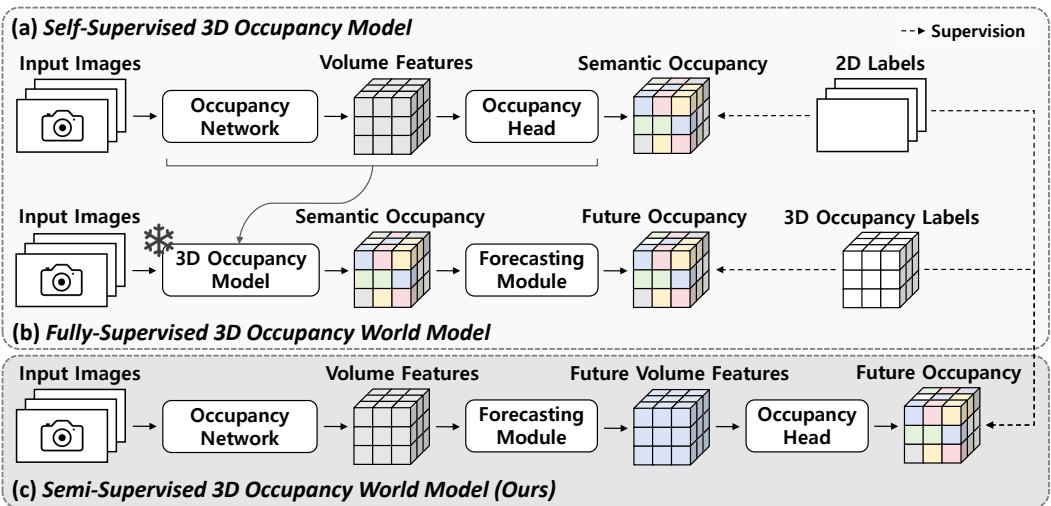

Figure 1: **(a) Self-Supervised 3D Occupancy Model** can be trained using solely 2D labels as supervision. However, it lacks the capability to forecast future occupancy. In contrast, **(b) Fully-Supervised 3D Occupancy World Model** can forecast future occupancy, but it relies on 3D occupancy labels for meaningful results due to its indirect architecture, which employs a frozen 3D occupancy model. To tackle these challenges, our **(c) Semi-Supervised 3D Occupancy World Model**, featuring 2D rendering supervision and an end-to-end architecture, can forecast future occupancy straightly from image inputs while taking advantage of 2D labels.

---

[*]Corresponding authors.
[1]Codes and models can be accessed at `https://github.com/getterupper/PreWorld`.

## 1 INTRODUCTION

3D scene understanding forms the cornerstone of autonomous driving, exerting a direct influence on downstream tasks such as planning and navigation. Among various 3D scene understanding tasks (Wang et al., 2022; Li et al., 2022a; Wei et al., 2023; Jin et al., 2024), *3D Occupancy Prediction* plays a crucial role in autonomous systems. Its objective is to predict the semantic occupancy of each voxel throughout the entire scene from limited observation. To this end, some previous methods (Liong et al., 2020; Cheng et al., 2021; Xia et al., 2023) prioritize LiDAR as input modality due to its robust performance in capturing accurate geometric information. Nevertheless, they are often considered hardware-expensive. Consequently, there has been a shift towards vision-centric solutions in recent years (Zhang et al., 2023c; Li et al., 2023a; Zheng et al., 2024).

Despite significant advancements in aforementioned methods, they primarily focus on enhancing better perception of the current scene. For advanced collision avoidance and route planning, autonomous vehicles need to not only comprehend the current scene but also forecast the evolution of future scenes based on the understanding of world dynamics. Therefore, *4D Occupancy Forecasting* has been introduced to forecast future 3D occupancy given historical observations. Recent works have aimed to achieve this by learning a 3D occupancy world model (Zheng et al., 2023; Wei et al., 2024). However, when processing image inputs, these methods follow an circuitous path, as shown in Fig 1 (b). Typically, a pre-trained 3D occupancy model is employed to obtain current occupancy, which is then fed into a forecasting module to generate future occupancy. The forecasting module includes a tokenizer that encodes occupancy into discrete tokens, an autoregressive architecture to generate future tokens, and a decoder to obtain future occupancy. Information loss is prone to occur in such repeated encoding and decoding processes. Hence, existing methods heavily rely on 3D occupancy labels as supervision to produce meaningful results, leading to notable annotation costs.

In contrast to 3D occupancy labels, 2D labels are relatively easier to acquire. Recently, employing purely 2D labels for self-supervised learning has shown some promising results in 3D occupancy prediction task, as illustrated in Fig 1 (a). By utilizing volumetric rendering, RenderOcc (Pan et al., 2024) employs 2D depth maps and semantic labels to train the model. Methods like SelfOcc (Huang et al., 2024) and OccNerf (Zhang et al., 2023a) take a step further, using only image sequences as supervision. However, there have not been similar attempts in 4D occupancy forecasting task.

Based on the above observations, we propose **PreWorld**, a semi-supervised vision-centric 3D occupancy world model, designed to fulfill the utility of 2D labels during training, while achieving competitive performance across both 3D occupancy prediction and 4D occupancy forecasting tasks, as shown in Fig 1 (c). To this end, we propose a novel two-stage training paradigm: the self-supervised pre-training stage and the fully-supervised fine-tuning stage. Inspired by RenderOcc, during the pre-training stage, we introduce an attribute projection head to obtain diverse attribute fields of current and future scenes (e.g., RGB, density, semantic), facilitating temporal supervision through 2D labels using volume rendering techniques. Moreover, we propose a simple yet effective state-conditioned forecasting module, which allows us to simultaneously optimize occupancy network and forecasting module, and directly forecast future 3D occupancy based on multi-view image inputs in an end-to-end manner, thus avoiding possible information loss.

To demonstrate the effectiveness of PreWorld, we conduct extensive experiments on the widely used Occ3D-nuScenes benchmark (Tian et al., 2024) and compare with recent methods using both 2D and/or 3D supervision. Experimental results indicate that our approach can yield competitive performance across multiple tasks. For 3D occupancy prediction, PreWorld outperforms the previous best method OccFlowNet (Boeder et al., 2024) with an mIoU of 34.69 over 33.86. For 4D occupancy forecasting, PreWorld sets the new SOTA performance, outperforming existing methods OccWorld (Zheng et al., 2023) and OccLLaMA (Wei et al., 2024). For motion planning, PreWorld yields comparable and often better results than other vision-centric methods (Hu et al., 2022; Jiang et al., 2023; Tong et al., 2023). Furthermore, we validate the scalability of our two-stage training paradigm, showcasing its potential for large-scale training.

Our main contributions are as follows:

- A semi-supervised vision-centric 3D occupancy world model, PreWorld, which takes advantage of both 2D labels and 3D occupancy labels during training.

- A novel two-stage training paradigm, the effectiveness and scalability of which has been validated by extensive experiments.

- A simple yet effective state-conditioned forecasting module, enabling simultaneous optimization with occupancy network and direct future forecasting based on visual inputs.

- Extensive experiments compared to SOTA method, demonstrating that our method achieves competitive performance across multiple tasks, including 3D occupancy prediction, 4D occupancy forecasting and motion planning.

## 2 RELATED WORK

### 2.1 3D OCCUPANCY PREDICTION

Due to its vital application in autonomous driving, 3D occupancy prediction has attracted considerable attention. According to the input modality, existing methods can be broadly categorized into LiDAR-based and vision-centric methods. While LiDAR-based methods excel in capturing geometric details (Tang et al., 2020; Ye et al., 2021; 2023), vision-centric methods have garnered growing interest in recent years due to their rich semantic information, cost-effectiveness, and ease of deployment (Philion & Fidler, 2020; Liu et al., 2023; Ma et al., 2024). However, these methods focus solely on understanding the current scene while ignoring the forecasting of future scene changes. Therefore in this paper, we follow the approach of OccWorld (Zheng et al., 2023) and endeavor to address both of these tasks in a unified manner.

### 2.2 WORLD MODELS FOR AUTONOMOUS DRIVING

The objective of world models is to forecast future scenes based on action and past observations (Ha & Schmidhuber, 2018). In autonomous driving, world models can be utilized to generate synthetic data and aid in decision making. Some previous approaches (Hu et al., 2023a; Gao et al., 2023; Wang et al., 2024) aim to generate image sequences of outdoor driving scenarios using large pretrained generative models. However, relying on 2D images as scene representations leads to the lack of structural information. Some works (Khurana et al., 2022; 2023; Zhang et al., 2023b) tend to generate 3D point clouds, which on the other hand, fail to capture the semantic of the scene.

Recent attempts have emerged to generate 3D occupancy representations, which combine an understanding of both semantic and geometric information. The pioneering OccWorld (Zheng et al., 2023) introduces the 3D occupancy world model that, employing an autoregressive architecture, can forecast future occupancy based on current observation. Taking it a step further, OccLLaMA (Wei et al., 2024) integrates occupancy, action, and language, enabling 3D occupancy world model to possess reasoning capabilities. However, when it comes to vision-centric approaches, they both adopt an indirect path, requiring the usage of pre-trained 3D occupancy models for current occupancy prediction, succeeded by an arduous encoding-decoding process to forecast future occupancy. This manner poses challenges in model training, thus necessitating 3D occupancy labels as supervision to yield effective results. Considering this, we explore a straightforward way to directly forecast future occupancy using image inputs.

### 2.3 SELF-SUPERVISED 3D OCCUPANCY PREDICTION

While 3D occupancy provides rich structural information for training, it necessitates expensive and laborious annotation processes. In contrast, 2D labels are more readily obtainable, presenting an opportunity for self-supervised 3D occupancy prediction. Recently, some works have explored using Neural Radiance Fields (NeRFs) (Mildenhall et al., 2021) to perform volume rendering of scenes, thereby enabling 2D supervision for the model. RenderOcc (Pan et al., 2024) tends to use 2D depth maps and semantic labels for training. Despite significant performance gaps compared to existing methods, SelfOcc (Huang et al., 2024) and OccNeRF (Zhang et al., 2023a) have made meaningful attempts, aiming to solely utilize image sequences for self-supervised learning.

On the contrary, self-supervised approaches have not yet been observed in the realm of 4D occupancy forecasting tasks. Although OccWorld (Zheng et al., 2023) offers a self-supervised setting, it merely relies on an existing self-supervised 3D occupancy model to produce current occupancy

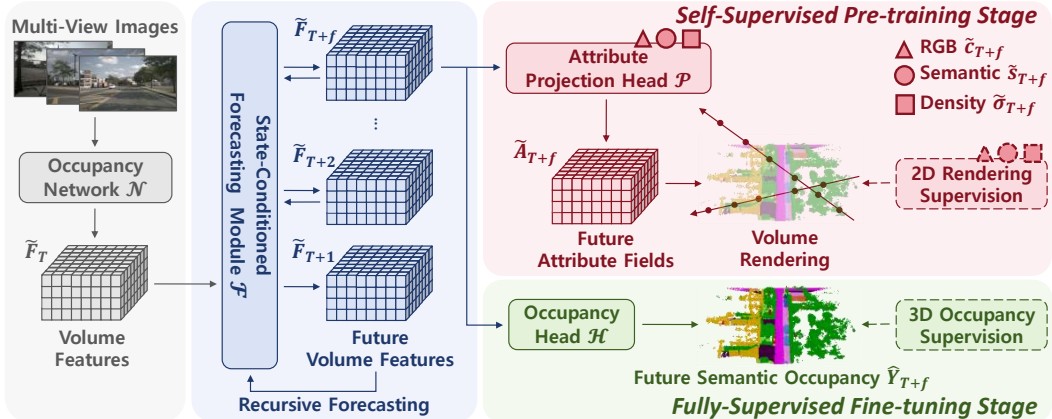

Figure 2: **The architecture of our proposed PreWorld.** Firstly, volume features are extracted from multi-view images with an occupancy network. Subsequently, a state-conditioned forecasting module is employed to recursively forecast future volume features using historical features. In the self-supervised pre-training stage, volume features are projected into various attribute fields and supervised by 2D labels through volume rendering techniques. In the fully-supervised fine-tuning stage, the attribute projection head no longer participates in the computations, occupancy predictions are directly obtained via an occupancy head and supervised by 3D occupancy labels.

without engaging in novel endeavors, and it also suffers from subpar performance. Different from OccWorld, we attempt to directly supervise future scenes using 2D labels, thereby optimizing our performance in both 3D occupancy prediction and 4D occupancy forecasting tasks simultaneously.

## 3 METHOD

### 3.1 REVISITING 4D OCCPUANCY FORECASTING

For the vehicle at timestamp $T$, vision-centric 3D occupancy prediction task takes $N$ views of images $S_T = \{I^1, I^2, ..., I^N\}$ as input and predicts current 3D occupancy $\hat{Y}_T \in \mathbb{R}^{X \times Y \times Z \times C}$ as output, where $(X, Y, Z)$ denote the resolution of the 3D volume and $C$ represents the number of semantic categories, including non-occupied (Huang et al., 2023; Zhang et al., 2023c; Liu et al., 2023; Pan et al., 2024). A 3D occupancy model $\mathbb{O}$ typically comprises an occupancy network $\mathcal{N}$ and an occupancy head $\mathcal{H}$. The process of occupancy prediction can be formulated as:

$$F_T = \mathcal{N}(S_T), \ \hat{Y}_T = \mathcal{H}(F_T), \tag{1}$$

where $\mathcal{N}$ extracts 3D volume features $F_T \in \mathbb{R}^{X \times Y \times Z \times D}$ from 2D image inputs ($D$ denotes the dimension of volume features), and $\mathcal{H}$ serves as a decoder to convert $F_T$ into 3D occupancy.

Vision-centric 4D occupancy forecasting task, on the other hand, utilizes an image sequence of past $k$ frames $\{S_T, S_{T-1}, ..., S_{T-k}\}$ as input, aiming at forecasting 3D occupancy of future $f$ frames (Zheng et al., 2023; Wei et al., 2024). A 3D occupancy world model $\mathbb{W}$ attempt to achieve this by adopting an auto-regressive manner:

$$\hat{Y}_{T+1} = \mathbb{W}(S_T, S_{T-1}, ..., S_{T-k}). \tag{2}$$

To this end, $\mathbb{W}$ employs an available 3D occupancy model $\mathbb{O}$ to predict 3D occupancy of past $k$ frames $\{\hat{Y}_T, ..., \hat{Y}_{T-k}\}$, and leverages a scene tokenizer $\mathcal{T}$, an autoregressive architecture $\mathcal{A}$ and a decoder $\mathcal{D}$ to forecast future 3D occupancy. After obtaining historial occupancy, $\mathbb{W}$ encodes 3D occupancy into discrete tokens $\{z_T, ..., z_{T-k}\}$ through $\mathcal{T}$. Subsequently, $\mathcal{A}$ is utilized to forecast future token $z_{T+1}$ based on these tokens, which is then input into $\mathcal{D}$ to generate future occupancy $\hat{Y}_{T+1}$. Formally, the process of occupancy forecasting can be formulated as follows:

$$\hat{Y}_T, ..., \hat{Y}_{T-k} = \mathbb{O}(S_T), ..., \mathbb{O}(S_{T-k}),$$
$$z_T, ..., z_{T-k} = \mathcal{T}(\hat{Y}_T), ..., \mathcal{T}(\hat{Y}_{T-k}), \tag{3}$$
$$z_{T+1} = \mathcal{A}(z_T, ..., z_{T-k}), \ \hat{Y}_{T+1} = \mathcal{D}(z_{T+1}).$$

Here, we need to mention that $\mathbb{O}$ is pre-trained and frozen during training. For example, Occ-World (Zheng et al., 2023) utilizes TPVFormer (Huang et al., 2023) as $\mathbb{O}$, while OccLLaMA (Wei et al., 2024) chooses FB-OCC (Li et al., 2023c).

## 3.2 STATE-CONDITIONED FORECASTING MODULE

Different from these approaches, we tend to a more straightforward path, which enables us to optimize 3D occupancy model and forecasting module simultaneously. Specially, we employ a state-conditioned forecasting module $\mathcal{F}$ instead of the combination of $\mathcal{T}$, $\mathcal{A}$ and $\mathcal{D}$, as illustrated in Fig 3. We formulate our approach of occupancy forecasting as follows:

$$\tilde{F}_T = \mathcal{N}(S_T, S_{T-1}, ..., S_{T-k}), \ \tilde{F}_{T+1} = \mathcal{F}(\tilde{F}_T), \ \hat{Y}_{T+1} = \mathcal{H}(\tilde{F}_{T+1}), \tag{4}$$

where we leverages $\mathcal{N}$ to extract volume features $\tilde{F}_T$ from temporal images, $\mathcal{F}$ to directly forecast future volume features $\tilde{F}_{T+1}$ and $\mathcal{H}$ to transform $\tilde{F}_{T+1}$ into future occupancy $\hat{Y}_{T+1}$.

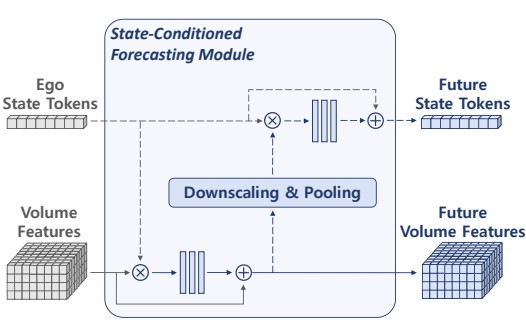

Figure 3: The proposed state-conditioned forecasting module is simply composed of two MLPs. Ego states can be optionally integrated into the network, as denoted by the dashed arrows.

Without loss of generality, our forecasting module is simply composed of two MLPs. We demonstrate that even without intricate design, this simple architecture can still achieve comparable and even superior results to state-of-the-art methods. This design showcases that previous practice of solely optimizing the forecasting module during training has its limitations. By simultaneously optimizing the occupancy network and forecasting module, 3D occupancy world models can achieve stronger performance. Additionally, our module can optionally incorporate ego-state information such as speed, acceleration and historical trajectories into the network. In Section 4.3, we demonstrate that this approach can further enhance the forecasting capabilities of the model.

Furthermore, this architecture brings an additional benefit for us. Given that previous forecasting modules encode scenes into discrete tokens, they cannot directly supervise future predictions with 2D labels via volume rendering, as done by self-supervised 3D occupancy models (Zhang et al., 2023a; Huang et al., 2024). Since our module preserves the volume features of future scenes, it provides an opportunity to train 3D occupancy world models in a self-supervised manner.

## 3.3 TEMPORAL 2D RENDERING SELF-SUPERVISION

**Attribute Projection.** Inspired by Pan et al. (2024), we transform the temporal volume feature sequence of current and future $f$ frames $\{\tilde{F}\}_t = \{\tilde{F}_T, \tilde{F}_{T+1}, ..., \tilde{F}_{T+f}\}$ into temporal attribute fields $\{\tilde{A}\}_t$ through an attribute projection head $\mathcal{P}$:

$$\{\tilde{A}\}_t = \{(\tilde{\sigma}, \tilde{s}, \tilde{c})\}_t = \mathcal{P}(\{\tilde{F}\}_t), \tag{5}$$

where $\tilde{\sigma} \in \mathbb{R}^{X \times Y \times Z \times 1}$, $\tilde{s} \in \mathbb{R}^{X \times Y \times Z \times D}$ and $\tilde{c} \in \mathbb{R}^{X \times Y \times Z \times 3}$ denote the density, semantic and RGB fields of the 3D volume, respectively. In implementation, $\mathcal{P}$ comprises several MLPs, which is validated to be a simple yet effective method (Boeder et al., 2024).

**Ray Generation.** Given the intrinsic and extrinsic parameters of camera $j$ at timestamp $i$, we can extract a set of 3D rays $\{r\}_i^j$, where each ray $r$ originates from camera $j$ and corresponds to a pixel of the image $I_i^j$. Additionally, we can utilize ego pose matrices to transform rays from adjacent $n$ frames to current frame, enabling better capture of surrounding information. These rays collectively constitute the set $\{r\}_i$ utilized for supervising $\tilde{A}_i = (\tilde{\sigma}_i, \tilde{s}_i, \tilde{c}_i)$.

**Volume Rendering.** For each $r \in \{r\}_i$, we sample $M$ points $\{u_m\}_{m=1}^M$ along the ray. Then the rendering weight $w(u_m)$ of each sampled point $u_m$ can be computed by:

$$T(u_m) = \exp(-\sum_{p=1}^{m-1} \tilde{\sigma}_i(u_p)\delta_p), \; w(u_m) = T(u_m)(1 - \exp(-\tilde{\sigma}_i(u_m)\delta_m)), \quad (6)$$

where $T(u_m)$ denotes the accumulated transmittance until $u_m$, and $\delta_m = u_{m+1} - u_m$ denotes the interval between adjacent sampled points. Finally, the 2D rendered depth, semantic and RGB predictions $(\hat{d}_i^{2D}(r), \hat{s}_i^{2D}(r), \hat{c}_i^{2D}(r))$ can be computed by cumulatively summing the products of the values corresponding to each point along the ray and their respective rendering weights:

$$\hat{d}_i^{2D}(r) = \sum_{m=1}^M w(u_m)u_m, \; \hat{s}_i^{2D}(r) = \sum_{m=1}^M w(u_m)\tilde{s}_i(u_m), \; \hat{c}_i^{2D}(r) = \sum_{m=1}^M w(u_m)\tilde{c}_i(u_m). \quad (7)$$

**Temporal 2D Rendering Supervision.** After acquiring 2D rendered predictions $(\hat{d}_i^{2D}, \hat{s}_i^{2D}, \hat{c}_i^{2D})$ with 3D ray set $\{r\}_i$, the temporal 2D rendering loss can be formulated as:

$$\mathcal{L}_{2D} = \sum_{i=T}^{T+f} \lambda_{dep}\mathcal{L}_{dep}(d_i^{2D}, \hat{d}_i^{2D}) + \lambda_{sem}\mathcal{L}_{sem}(s_i^{2D}, \hat{s}_i^{2D}) + \lambda_{RGB}\mathcal{L}_{RGB}(c_i^{2D}, \hat{c}_i^{2D}), \quad (8)$$

where $(d_i^{2D}, s_i^{2D}, c_i^{2D})$ represents 2D depth map, semantic label and RGB of corresponding pixels.

### 3.4 Two-Stage Training Paradigm

**Training Scheme.** As illustrated in Fig 2, our training scheme for PreWorld includes two stages: In the self-supervised pre-training stage, as illustrated in Section 3.3, we employs the attribute projection head $\mathcal{P}$ to enable temporal supervision with 2D labels. This approach allows us to leverage the abundant and easily obtainable 2D labels, while preemptively optimizing both the occupancy network $\mathcal{N}$ and forecasting module $\mathcal{F}$. In the subsequent fine-tuning stage, we utilize a occupancy head $\mathcal{H}$ to produce occupancy results and use 3D occupancy labels for further optimization.

**Training Loss.** For pre-training stage, we employ temporal 2D rendering loss $\mathcal{L}_{2D}$ as formulated in Eq. 8. Specially, we utilize SILog loss and cross-entropy loss from Pan et al. (2024) as $\mathcal{L}_{dep}$ and $\mathcal{L}_{sem}$, respectively, and use L1 loss as $\mathcal{L}_{RGB}$. For fine-tuning stage, we employ focal loss $\mathcal{L}_f$, lovasz-softmax loss $\mathcal{L}_l$ and scene-class affinity loss $\mathcal{L}_{scal}^{sem}$ and $\mathcal{L}_{scal}^{geo}$, following the practice of Li et al. (2023c). Therefore, the total loss function for fine-tuning stage can be represented as follows:

$$\mathcal{L}_{3D} = \lambda_f\mathcal{L}_f + \lambda_l\mathcal{L}_l + \lambda_{scal}^{sem}\mathcal{L}_{scal}^{sem} + \lambda_{scal}^{geo}\mathcal{L}_{scal}^{geo}. \quad (9)$$

## 4 Experiments

### 4.1 Experiment Settings

**Dataset and Metrics.** Our experiments are conducted on the Occ3D-nuScenes benchmark (Tian et al., 2024), which provides dense semantic occupancy annotations for the widely used nuScenes dataset (Caesar et al., 2020). Each annotation covers a range of $[-40 \sim 40m, -40 \sim 40m, -1 \sim 5.4m]$ around the ego vehicle. The ground-truth semantic occupancy is represented as $200 \times 200 \times 16$ 3D voxel grids with $0.4$m resolution. Each voxel is annotated with 18 classes (17 semantic classes and 1 free). The official split for training and validation sets is employed. Following common practices, we use mIoU and IoU as the evaluation metric for 3D occupancy prediction and 4D occupancy forecasting tasks, and use L2 error and collision rate for motion planning task.

**Implementation Details.** We use the identical network architecture for all the three tasks, yet for the non-temporal 3D occupancy prediction task, we omit temporal supervision and losses accordingly. We adopt BEVStereo (Li et al., 2023b) as the occupancy network $\mathcal{N}$, only replacing its detection head with the occupancy head $\mathcal{H}$ from FB-OCC Li et al. (2023c) to produce occupancy prediction. For training, we set the batch size to 16, use Adam as the optimizer, and train with a

Table 1: **3D occupancy prediction performance on the Occ3D-nuScenes dataset.** GT represents the type of labels used during training. The best and second-best performances are represented by **bold** and underline respectively.

| Method | GT | others | barrier | bicycle | bus | car | cons. veh | motorcycle | pedestrian | traffic cone | trailer | truck | dri. sur | other flat | sidewalk | terrain | mammade | vegetation | mIoU (%) |
|---|---|---|---|---|---|---|---|---|---|---|---|---|---|---|---|---|---|---|---|
| SelfOcc (Huang et al., 2024) | 2D | 0.00 | 0.15 | 0.66 | 5.46 | 12.54 | 0.00 | 0.80 | 2.10 | 0.00 | 0.00 | 8.25 | 55.49 | 0.00 | 26.30 | 26.54 | 14.22 | 5.60 | 9.30 |
| OccNeRF (Zhang et al., 2023a) | 2D | 0.00 | 0.83 | 0.82 | 5.13 | 12.49 | 3.50 | 0.23 | 3.10 | 1.84 | 0.52 | 3.90 | 52.62 | 0.00 | 20.81 | 24.75 | 18.45 | 13.19 | 9.53 |
| RenderOcc (Pan et al., 2024) | 2D | 5.69 | 27.56 | 14.36 | 19.91 | 20.56 | 11.96 | 12.42 | 12.14 | 14.34 | 20.81 | 18.94 | **68.85** | 33.35 | 42.01 | 43.94 | 17.36 | 22.61 | 23.93 |
| OccFlowNet (Boeder et al., 2024) | 2D | 1.60 | 27.50 | 26.00 | 34.00 | 32.00 | 20.40 | 25.90 | 18.60 | 20.20 | 26.00 | 28.70 | 62.00 | 27.20 | 37.80 | 39.50 | 29.00 | 26.80 | 28.42 |
| MonoScene (Cao & De Charette, 2022) | 3D | 1.75 | 7.23 | 4.26 | 4.93 | 9.38 | 5.67 | 3.98 | 3.01 | 5.90 | 4.45 | 7.17 | 14.91 | 6.32 | 7.92 | 7.43 | 1.01 | 7.65 | 6.06 |
| TPVFormer (Huang et al., 2023) | 3D | 7.22 | 38.90 | 13.67 | 40.78 | 45.90 | 17.23 | 19.99 | 18.85 | 14.30 | 26.69 | 34.17 | 55.65 | 35.47 | 37.55 | 30.70 | 19.40 | 16.78 | 27.83 |
| BEVDet (Huang et al., 2021) | 3D | 4.39 | 30.31 | 0.23 | 32.26 | 34.47 | 12.97 | 10.34 | 10.36 | 6.26 | 8.93 | 23.65 | 52.27 | 24.61 | 26.06 | 22.31 | 15.04 | 15.10 | 19.38 |
| OccFormer (Zhang et al., 2023c) | 3D | 5.94 | 30.29 | 12.32 | 34.40 | 39.17 | 14.44 | 16.45 | 17.22 | 9.27 | 13.90 | 26.36 | 50.99 | 30.96 | 34.66 | 22.73 | 6.76 | 6.97 | 21.93 |
| BEVFormer (Li et al., 2022b) | 3D | 5.85 | 37.83 | 17.87 | 40.44 | 42.43 | 7.36 | 23.88 | 21.81 | 20.98 | 22.38 | 30.70 | 55.35 | 28.36 | 36.00 | 28.06 | 20.04 | 17.69 | 26.88 |
| RenderOcc (Pan et al., 2024) | 2D+3D | 4.84 | 31.72 | 10.72 | 27.67 | 26.45 | 13.87 | 18.20 | 17.67 | 17.84 | 21.19 | 23.25 | 63.20 | 36.42 | **46.21** | **44.26** | 19.58 | 20.72 | 26.11 |
| CTF-Occ (Tian et al., 2024) | 3D | 8.09 | 39.33 | 20.56 | 38.29 | 42.24 | 16.93 | 24.52 | 22.72 | 21.05 | 22.98 | 31.11 | 53.33 | 33.84 | 37.98 | 33.23 | 20.79 | 18.00 | 28.53 |
| SparseOcc (Liu et al., 2023) | 3D | - | - | - | - | - | - | - | - | - | - | - | - | - | - | - | - | - | 30.90 |
| OccFlowNet (Boeder et al., 2024) | 2D+3D | 8.00 | 37.60 | 26.00 | 42.10 | 42.50 | 21.60 | 29.20 | 22.30 | 25.70 | 29.70 | 34.40 | 64.90 | 37.20 | 44.30 | 43.20 | **34.30** | 32.50 | 33.86 |
| **PreWorld (Ours)** | 3D | 10.83 | 44.13 | **26.35** | 42.16 | 46.15 | 22.92 | 28.86 | 26.89 | 26.44 | 28.29 | 34.43 | 65.67 | 35.91 | 41.09 | 37.41 | 30.16 | 29.54 | 33.95 |
| **+ Pre-training** | 2D+3D | **11.81** | **45.01** | 26.29 | **43.32** | **47.71** | 24.23 | 31.29 | 27.41 | 27.68 | 30.62 | 35.64 | 63.71 | 37.27 | 41.20 | 37.54 | 29.36 | 29.70 | **34.69** |

learning rate of $1 \times 10^{-4}$. All the hyperparameters $\lambda$ in the loss functions have been set to 1.0. For 3D occupancy prediction task, PreWorld undergoes 6 epochs in self-supervised pre-training stage and 12 epochs in fully-supervised fine-tuning stage. For 4D occupancy forecasting and motion planning task, PreWorld undergoes 8 epochs in self-supervised pre-training stage and 18 epochs in fully-supervised fine-tuning stage. All experiments are conducted on 8 NVIDIA A100 GPUs.

## 4.2 RESULTS AND ANALYSIS

Table 2: **4D occupancy forecasting performance on the Occ3D-nuScenes dataset.** The latest vision-centric approaches of OccWorld (Zheng et al., 2023) and OccLLaMA (Wei et al., 2024) are taken as baselines for fair comparison. Aux. Sup. represents auxiliary supervision apart from the ego trajectory. Avg. reprersents the average performance of that in 1s, 2s, and 3s. The best and second-best performances are represented by **bold** and underline respectively.

| Method | Aux. Sup. | mIoU (%) ↑ | | | | IoU (%) ↑ | | | |
|---|---|---|---|---|---|---|---|---|---|
| | | 1s | 2s | 3s | Avg. | 1s | 2s | 3s | Avg. |
| OccWorld-S | None | 0.28 | 0.26 | 0.24 | 0.26 | 5.05 | 5.01 | 4.95 | 5.00 |
| OccWorld-T | Semantic LiDAR | 4.68 | 3.36 | 2.63 | 3.56 | 9.32 | 8.23 | 7.47 | 8.34 |
| OccWorld-D | 3D Occ | 11.55 | 8.10 | 6.22 | 8.62 | 18.90 | 16.26 | 14.43 | 16.53 |
| OccLLaMA-F | 3D Occ | 10.34 | 8.66 | 6.98 | 8.66 | **25.81** | **23.19** | **19.97** | **22.99** |
| **PreWorld (Ours)** | 3D Occ | 11.69 | 8.72 | 6.77 | 9.06 | 23.01 | 20.79 | 18.84 | 20.88 |
| **+ Pre-training** | 2D Labels & 3D Occ | **12.27** | **9.24** | **7.15** | **9.55** | 23.62 | 21.62 | 19.63 | 21.62 |

**3D Occupancy Prediction.** We first compare the 3D occupancy prediction performance of our PreWorld model with the latest methods on the Occ3D-nuScenes dataset. As shown in Table 1, PreWorld achieves an mIoU of 34.69, surpassing the previous state-of-the-art method, OccFlowNet (Boeder et al., 2024), which has an mIoU of 33.86, as well as other methods using 2D, 3D, or combined supervision. This highlights the effectiveness of PreWorld in perceiving the current scene. Additionally, the proposed 2D pre-training stage boosts performance by 0.74 mIoU, with improvements observed across nearly all categories, both static and dynamic. These results underscore the importance of the proposed 2D pre-training stage for enhanced scene understanding.

In Figure 4, we further compare the qualitative results of PreWorld with the latest fully-supervised method SparseOcc (Liu et al., 2023) and self-supervised method RenderOcc (Pan et al., 2024). RenderOcc can project scene voxels onto multi-view images to obtain comprehensive supervision from various ray directions, thus capturing abundant geometric and semantic information from 2D labels. However, as shown in the last column, it struggles in predicting unseen regions and understanding the overall scene structure. On the other hand, SparseOcc excels in predicting scene structures. Yet owing to insufficient supervision for small objects and long-tailed objects from 3D occupancy labels, it often encounters information loss when predicting objects like *poles* and *motorcycles*, as

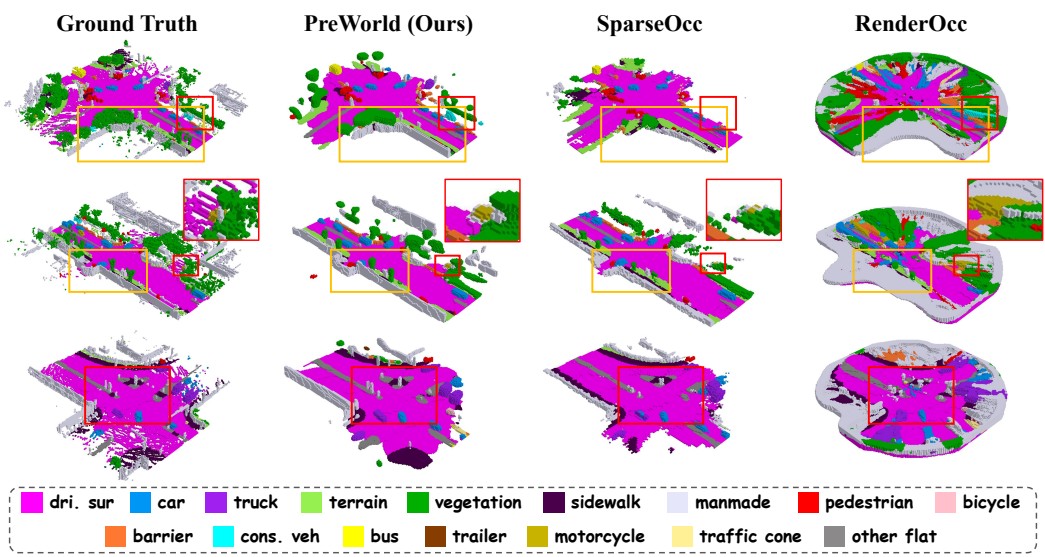

Figure 4: **Qualitative results of 3D occupancy prediction on the Occ3D-nuScenes validation set.** The holistic structure and fine-grained details of the scene are highlighted by orange boxes and red boxes respectively. Compared with existing fully-supervised methods and self-supervised methods, PreWorld can obtain better scene structure and capture finer local details.

Table 3: **Motion planning performance on the Occ3D-nuScenes dataset.** The latest vision-centric approaches of OccWorld (Zheng et al., 2023) and OccLLaMA (Wei et al., 2024) are taken as baselines for fair comparison. † represents training and inference with ego state information introduced. The best and second-best performances are represented by **bold** and underline respectively.

| Method | Aux. Sup. | L2 (m) ↓ | | | | Collision Rate (%) ↓ | | | |
|---|---|---|---|---|---|---|---|---|---|
| | | 1s | 2s | 3s | Avg. | 1s | 2s | 3s | Avg. |
| ST-P3 (Hu et al., 2022) | Map & Box & Depth | 1.33 | 2.11 | 2.90 | 2.11 | 0.23 | 0.62 | 1.27 | 0.71 |
| UniAD (Hu et al., 2023b) | Map & Box & Motion & Track & 3D Occ | 0.48 | 0.96 | 1.65 | 1.03 | 0.05 | **0.17** | **0.71** | **0.31** |
| VAD (Jiang et al., 2023) | Map & Box & Motion | 0.54 | 1.15 | 1.98 | 1.22 | **0.04** | 0.39 | 1.17 | 0.53 |
| OccNet (Tong et al., 2023) | Map & Box & 3D Occ | 1.29 | 2.13 | 2.99 | 2.14 | 0.21 | 0.59 | 1.37 | 0.72 |
| OccWorld-S† | None | 0.67 | 1.69 | 3.13 | 1.83 | 0.19 | 1.28 | 4.59 | 2.02 |
| OccWorld-T† | Semantic LiDAR | 0.54 | 1.36 | 2.66 | 1.52 | 0.12 | 0.40 | 1.59 | 0.70 |
| OccWorld-D† | 3D Occ | 0.52 | 1.27 | 2.41 | 1.40 | 0.12 | 0.40 | 2.08 | 0.87 |
| OccLLaMA-F† | 3D Occ | 0.38 | 1.07 | 2.15 | 1.20 | 0.06 | 0.39 | 1.65 | 0.70 |
| **PreWorld (Ours)** | 3D Occ | 0.49 | 1.22 | 2.32 | 1.34 | 0.19 | 0.57 | 2.65 | 1.14 |
| **+ Pre-training** | 2D Label & 3D Occ | 0.41 | 1.16 | 2.32 | 1.30 | 0.50 | 0.88 | 2.42 | 1.27 |
| **PreWorld (Ours)†** | 3D Occ | **0.22** | 0.31 | 0.41 | **0.31** | 0.36 | 0.52 | 0.73 | 0.54 |
| **+ Pre-training†** | 2D Label & 3D Occ | **0.22** | **0.30** | **0.40** | **0.31** | 0.21 | 0.66 | **0.71** | 0.53 |

shown in the second and the last row. In contrast, our model is initially pre-trained with 2D labels, thereby gaining a sufficient understanding of the scene geometry and semantics. In the fine-tuning stage, the model is further optimized using 3D occupancy labels, enabling PreWorld to better predict scene structures. Consequently, PreWorld performs comparably to SparseOcc in holistic structure predictions but exhibits a clear advantage in predicting fine-grained local details, underscoring the superiority of our training paradigm.

**4D Occupancy Forecasting.** Table 2 presents the 4D occupancy forecasting performance of PreWorld compared to existing baseline models, OccWorld(Zheng et al., 2023) and OccLLaMA(Wei et al., 2024). When using only 3D occupancy supervision, our method achieves the highest mIoU over the future 3-second interval, outperforming the baselines. This demonstrates the effectiveness of our cooperative training approach for both occupancy feature extraction and forecasting modules

in an end-to-end manner. Similar to the results for 3D occupancy prediction, incorporating the 2D pre-training stage further improves both mIoU and IoU across all future timestamps. This highlights how pre-training provides valuable geometric and semantic auxiliary information from dense 2D image representations. Given that 2D labels are more readily available than costly 3D occupancy annotations, the performance boost from the two-stage training paradigm of PreWorld is noteworthy.

**Motion Planning.** The motion planning results are further compared in Table 3. Without incorporating ego-state information, our model performs comparably to occupancy world models and even some well-designed planning models. When ego-state information is utilized following the same configuration as OccWorld and OccLLaMA (indicated in gray), our method achieves SOTA performance with significant improvements, further enhanced by the pre-training stage. Since PreWorld follows a direct training paradigm, taking the original images as input and producing planning results, the impact of ego-state is notably different from that in world model baselines. We attribute this difference to the "shortcut" effect observed in prior work (Zhai et al., 2023; Li et al., 2024). We leave the detailed analysis of the relationship between input ego-state, forecasted occupancy, and planning outcomes for future investigation.

## 4.3 ABLATION STUDY

Table 4: Ablation study of different supervision attributes utilized in pre-training stage.

| RGB | Depth | Semantic | mIoU (%) ↑ |
|-----|-------|----------|------------|
|     |       |          | 33.95 |
| ✓   |       |          | 34.11 (+0.16) |
| ✓   | ✓     |          | 34.43 (+0.48) |
| ✓   | ✓     | ✓        | **34.69 (+0.74)** |

Table 5: Ablation study of different data scale utilized in pre-training and fine-tuning stage.

| Fine-tuning | Pre-training | mIoU (%) ↑ |
|-------------|--------------|------------|
| 150 Scenes | ×
700 Scenes | 18.66
25.02 (+6.36) |
| 450 Scenes | ×
700 Scenes | 31.99
33.37 (+1.38) |
| 700 Scenes | ×
450 Scenes
700 Scenes | 33.95
34.28 (+0.33)
**34.69 (+0.74)** |

**Effectiveness of Pre-training.** The effectiveness of different supervision attributes of the 2D pre-training stage is analyzed in this section. As noted earlier, the benefits of pre-training are consistent across both 3D occupancy prediction and 4D occupancy forecasting. Therefore, to conserve computational resources, we perform ablation experiments on the 3D occupancy prediction task. Table 4 shows that as RGB, depth, and semantic attributes are progressively added during the pre-training stage, the final mIoU results steadily improve. This demonstrates the effectiveness of the three 2D supervision attributes, with even the simplest RGB attribute providing a boost in performance.

**Scalability of Pre-training.** To validate the scalability of our approach, we conduct ablation studies on the data scale used in both pre-training and fine-tuning stages, as shown in Table 5. Firstly, the introduction of the pre-training stage consistently improves performance across all fine-tuning data scales, where larger pre-training scale leads to better results. Secondly, when the fine-tuning dataset is small (150 scenes), which means costly 3D occupancy labels are limited, the pre-training stage significantly boosts the mIoU from 18.66 to 25.02. Thirdly, with pre-training, the model fine-tuned on a smaller dataset (450 scenes) achieves comparable performance to a model without pre-training but fine-tuned on a larger dataset (700 scenes), with mIoU of 33.37 and 33.95, respectively. These results highlight the effectiveness and scalability of our two-stage training paradigm.

**Model Components.** We perform ablation studies on the effectiveness of various components in our approach for 4D occupancy forecasting, as shown in Table 6. For comparison, we first present a Copy&Paste baseline, which simply copys the current occupancy prediction results of our best 3D occupancy prediction model and calculates the mIoU between these results and the ground truth of the future frames. This serves as a lower bound for PreWorld, showcasing the performance of a model without any future forecasting capabilities. The results in row 1 and row 2 demonstrate that our proposed forecasting module has effectively equipped the model with future forecasting capabilities. By introducing this straightforward design, the model can produce non-trivial results

Table 6: **Ablation study of different components in our approach.** The Copy&Paste employs our best model for 3D occupancy prediction task. Ego denotes using ego-state information during training. SSP denotes self-supervised pre-training for model. TS denotes trajectory supervision.

| Method | Ego | SSP | TS | mIoU (%) ↑ | | | | IoU (%) ↑ | | | |
|--------|-----|-----|-----|-----|-----|-----|-----|-----|-----|-----|-----|
| | | | | 1s | 2s | 3s | Avg. | 1s | 2s | 3s | Avg. |
| Copy&Paste | | | | 9.76 | 7.37 | 6.23 | 7.79 | 20.44 | 17.73 | 16.20 | 18.12 |
| PreWorld | | | | 11.12 | 7.73 | 5.89 | 8.25 (+0.46) | 22.91 | 20.31 | 17.84 | 20.35 (+2.23) |
| | ✓ | | | 11.17 | 8.54 | 6.83 | 8.85 (+1.06) | 23.27 | 20.83 | 18.51 | 20.87 (+2.75) |
| | ✓ | | ✓ | 11.69 | 8.72 | 6.77 | 9.06 (+1.27) | 23.01 | 20.79 | 18.84 | 20.88 (+2.76) |
| | ✓ | ✓ | | 11.58 | 9.14 | **7.34** | 9.35 (+1.56) | 23.27 | 21.41 | 19.49 | 21.39 (+3.27) |
| | ✓ | ✓ | ✓ | **12.27** | **9.24** | 7.15 | **9.55 (+1.76)** | **23.62** | **21.62** | **19.63** | **21.62 (+3.50)** |

Table 7: **Ablation study of joint training.** All results in the table are obtained utilizing ego-state information. Traj, 2D and 3D denote ego trajectory, 2D labels and 3D occupancy labels, respectively.

| Supervision | | | L2 (m) ↓ | | | | Collision Rate (%) ↓ | | | |
|------|-----|-----|-----|-----|-----|-----|-----|-----|-----|-----|
| Traj | 2D | 3D | 1s | 2s | 3s | Avg. | 1s | 2s | 3s | Avg. |
| ✓ | | | **0.20** | 0.34 | 0.80 | 0.45 | 0.50 | **0.62** | 0.90 | 0.67 |
| ✓ | | ✓ | 0.22 | 0.31 | 0.41 | **0.31** | 0.36 | 0.52 | 0.73 | 0.54 |
| ✓ | ✓ | ✓ | 0.22 | **0.30** | **0.40** | **0.31** | **0.21** | 0.66 | **0.71** | **0.53** |

and achieve significant performance enhancements, particularly evident in the IoU metric. Additionally, incorporating ego-state information and employing self-supervised pre-training further enhance both mIoU and IoU, as shown in row 3 and row 5. These findings underscore the importance and contribution of each component in our approach.

**Joint Training.** We further demonstrate the effectiveness of joint training. As shown in the row 4 and row 6 of Table 6, when simultaneously optimizing both 4D occupancy forecasting and motion planning tasks, the forecasting capabilities of PreWorld are further enhanced. The introduction of trajectory supervision has improved model performance regardless of the utilization of self-supervised pre-training, with an increase from 8.85 and 9.35 to 9.06 and 9.55 in mIoU, respectively. Furthermore, joint training has also enhanced the planning capabilities of our model. As shown in Table 7, compared to the model supervised solely by ego trajectory, model supervised using both ego trajectory and 3D occupancy labels exhibits a significant improvement in both L2 error and collision rates, while the introduction of 2D labels further elevates the model performance. These results collectively demonstrate that jointly training 4D occupancy forecasting and motion planning tasks, as opposed to training them separately, provides additional performance benefits for the model.

## 5 CONCLUSION

In this paper, we propose PreWorld, a semi-supervised vision-centric 3D occupancy world model for autonomous driving. We propose a novel two-stage training paradigm that allows our method to leverage abundant and easily accessible 2D labels for self-supervised pre-training. In the subsequent fine-tuning stage, the model is further optimized using 3D occupancy labels. Furthermore, we introduce a simple yet effective state-conditioned forecasting module, which addresses the challenge faced by existing methods in simultaneously optimizing the occupancy network and forecasting module. This module reduces information loss during training, while enabling the model to directly forecast future scenes and ego trajectory based on visual inputs. Through extensive experiments, we demonstrate the robustness of PreWorld across 3D occupancy prediction, 4D occupancy forecasting and motion planning tasks. Particularly, we validate the effectiveness and scalability of our training paradigm, outlining a viable path for scalable model training in autonomous driving scenarios.

ACKNOWLEDGMENTS

This project is supported by National Science and Technology Major Project (2022ZD0115502), Lenovo Research, and Wuxi Research Institute of Applied Technologies, Tsinghua University under Grant 20242001120.

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

## A MORE EVALUATIONS

### A.1 3D OCCUPANCY PREDICTION WITH RAYIOU

To address the inconsistent depth penalty issue within the mIoU metric, SparseOcc (Liu et al., 2023) introduces a novel metric, RayIoU, designed to enhance the evaluation of 3D occupancy model performance. In order to demonstrate the robustness of our approach as a 3D occupancy model across various metrics, we opt to evaluate PreWorld on the 3D occupancy prediction task using RayIoU as metric, and compare the results with existing methods in this section.

Table 8: **3D occupancy prediction performance on the Occ3D-nuScenes dataset.** We use RayIoU as the evaluation metric (Liu et al., 2023). The best and second-best performances are represented by **bold** and underline respectively.

| Method | RayIoU | RayIoU$_{1m, 2m, 4m}$ | | |
|---|---|---|---|---|
| BEVFormer(4f) | 32.4 | 26.1 | 32.9 | 38.0 |
| RenderOcc | 19.5 | 13.4 | 19.6 | 25.5 |
| SimpleOcc | 22.5 | 17.0 | 22.7 | 27.9 |
| BEVDet-Occ (2f) | 29.6 | 23.6 | 30.0 | 35.1 |
| BEVDet-Occ-Long (8f) | 32.6 | 26.6 | 33.1 | 38.2 |
| OccFlowNet | 32.6 | 25.6 | 33.3 | 38.8 |
| FB-OCC (16f) | 33.5 | 26.7 | 34.1 | 39.7 |
| SparseOcc (8f) | 34.0 | 28.0 | 34.7 | 39.4 |
| SparseOcc (16f) | 36.1 | 30.2 | 36.8 | 41.2 |
| **PreWorld (Ours)** | 36.4 | 30.0 | 37.2 | 41.9 |
| **+ Pre-training** | **38.7** | **32.5** | **39.6** | **44.0** |

As shown in Table 8, PreWorld achieves a RayIoU of 38.7, outperforming the previous SOTA method SparseOcc (Liu et al., 2023) by 2.6 RayIoU. Comparing to purely 3D occupancy supervision, the proposed self-supervised pre-training stage provides a significant boost in RayIoU from 36.4 to 38.7, which reaffirms the effectiveness of our two-stage training paradigm for PreWorld. Altogether, Table 1 and 8 showcases the strong performance of PreWorld across various metrics.

More importantly, the introduction of RayIoU explains the reason why our PreWorld does not outperform the baseline in certain categories. As shown in Table 1, these situations are predominantly focused on large static categories. For instance, in categories like *manmade* and *sidewalk*, the performance of PreWorld is surpassed by RenderOcc (Pan et al., 2024). SparseOcc points out that common practice in mIoU computation involves the utilization of visible masks, which only accounts for voxels within the visible region, without penalizing predictions outside this area. Consequently, many models can achieve higher mIoU scores by predicting thicker surfaces for large static categories. As demonstrated in the last column of Figure 4, RenderOcc, despite lacking an understanding of the overall scene structure, manages to attain higher scores in these categories through this strategy.

On the contrary, due to RayIoU considering the distance between voxels and the ego vehicle during computation, the model cannot gain an advantage by predicting thicker surfaces under this evaluation metric. Therefore, we believe that RayIoU is a more reasonable metric for comparing model performance in predicting large static categories. As shown in Table 8, when using RayIoU as the evaluation metric, the scores of both RenderOcc and OccFlowNet (Boeder et al., 2024) have decreased. While OccFlowNet outperforms SparseOcc in the mIoU metric with 33.86 over 30.90, its performance notably lags behind SparseOcc in terms of RayIoU. These results indicate that the performance of our PreWorld is not inferior in some categories; rather, our model tends to generate more reasonable predictions, which can be reflected in the RayIoU metric.

Likewise, we can explain why pre-training leads to a decline in model performance in certain categories. It can be observed that these instances also primarily focus on large static categories. For example, after pre-training, there is a significant mIoU performance decline in the *driveable surface* category. Based on the previous analysis, we showcase the RayIoU performance for the models on large static categories with and without pre-training.

Table 9: **Detailed 3D occupancy prediction performance of the large static categories on the Occ3D-nuScenes dataset.** We use RayIoU as the evaluation metric (Liu et al., 2023). GT represents the type of labels used during training. The best performances are represented by **bold**.

| Metric | Method | GT | RayIoU | barrier | Dri. Sur | other flat | sidewalk | terrain | manmade | vegetation |
|---|---|---|---|---|---|---|---|---|---|---|
| $RayIoU_{1m}$ | PreWorld | 3D | 30.0 | 39.4 | 56.4 | 24.2 | 25.5 | 23.8 | 32.9 | 23.2 |
| | + Pre-training | 2D+3D | **32.5** | **40.1** | **57.8** | **29.8** | **27.2** | **27.3** | **35.1** | **25.9** |
| $RayIoU_{2m}$ | PreWorld | 3D | 37.2 | 44.6 | 64.4 | 29.2 | 30.7 | 31.3 | 43.4 | 36.0 |
| | + Pre-training | 2D+3D | **39.6** | **45.4** | **65.9** | **34.3** | **32.7** | **34.6** | **44.7** | **37.8** |
| $RayIoU_{4m}$ | PreWorld | 3D | 41.9 | 46.6 | 72.4 | 33.2 | 35.6 | 38.1 | 49.7 | **47.3** |
| | + Pre-training | 2D+3D | **44.0** | **47.4** | **74.3** | **38.3** | **37.7** | **41.1** | **50.5** | 47.2 |

As shown in Table 9, the pre-trained model surpasses the model without pre-training on almost all large static categories across all thresholds. The results under the RayIoU metric indicate that pre-training steers the model towards predicting more plausible scene structures, rather than leading to performance decline. In conclusion, we believe that the results under RayIoU metric validate the effectiveness of pre-training and better showcase the robust prediction capabilities of our PreWorld.

## A.2 HOW PRE-TRAINING WORKS?

Due to time constraints, when conducting experiments on smaller datasets in Table 4, models fine-tuned on 150 scenes and 450 scenes are trained for 24 and 18 epochs, respectively, while the model on the full dataset is trained for 12 epochs. Considering the ratio of data reduction to extended training time, we believe that we do not allocate sufficient additional training time for experiments on the smaller datasets. Therefore in this section, to delve into how pre-training benefits the model, we extend the training duration across various settings to obtain more comprehensive experimental results, as presented in Table 10.

Table 10: **The extended ablation study of different data scale utilized in pre-training and fine-tuning stage.** The best performances are represented by **bold**.

| Fine-tuning | Pre-training | Epoch | | | | | |
|---|---|---|---|---|---|---|---|
| | | 12 | 18 | 24 | 36 | 48 | 60 |
| 150 Scenes | × | 11.18 | 13.85 | 18.66 | 29.30 | **30.26** | 30.00 |
| | 700 Scenes | 13.01 | 21.83 | 25.02 | **31.65** | 31.56 | 30.98 |
| 450 Scenes | × | 25.54 | 31.99 | 32.89 | **33.32** | - | - |
| | 700 Scenes | 29.52 | 33.37 | **34.19** | 34.08 | - | - |
| 700 Scenes | × | 33.95 | **33.99** | - | - | - | - |
| | 450 Scenes | **34.28** | 34.15 | - | - | - | - |
| | 700 Scenes | 34.69 | **34.89** | - | - | - | - |

As shown in the results, we believe that pre-training has benefited the model in two key aspects: on one hand, pre-training accelerates the convergence of the model; on the other hand, pre-training continues to enhance the model performance after convergence, thereby improving the data efficiency. Taking models fine-tuned on 150 scenes as an example, it can be observed that during the first 24 epochs, employing pre-training accelerates the convergence. Subsequently, both models have converged, with the pre-trained model still maintaining an advantage in prediction performance.

Furthermore, it can be observed that pre-training leads to a 0.87 mIoU improvement for the model fine-tuned on 450 scenes, while it results in a 0.90 mIoU improvement for the model fine-tuned on 700 scenes. We believe this situation is still related to the reasons analyzed in Section A.1, that is, for large static categories, existing evaluation metric does not adequately reflect the actual performance of the model. Therefore, we have detailed the corresponding mIoU for large static categories and small objects in Table 11.

Table 11: Detailed 3D occupancy prediction performance of different data scale utilized in pre-training and fine-tuning stage.

| Fine-tuning | Pre-training | mIoU | | |
| --- | --- | --- | --- | --- |
| | | Overall | Large Static | Small |
| 150 Scenes | × | 30.26 | 35.82 | 24.08 |
| | 700 Scenes | 31.65 (+1.39) | 37.18 (+1.36) | 25.50 (+1.42) |
| 450 Scenes | × | 33.32 | 38.40 | 26.29 |
| | 700 Scenes | 34.19 (+0.87) | 39.04 (+0.64) | 27.44 (+1.15) |
| 700 Scenes | × | 33.99 | 39.83 | 26.98 |
| | 700 Scenes | 34.89 (+0.90) | 40.72 (+0.89) | 27.96 (+0.98) |

It can be observed that the mIoU for large categories does not always effectively reflect the performance improvement of the model. For the model fine-tuned with 450 scenes, pre-training leads to a 0.64 increase in mIoU for large categories, while the model fine-tuned with 700 scenes sees an increase of 0.89. In contrast, the increase in mIoU for small objects can better reflect the effectiveness of pre-training, aligning with the expectations: 2D pre-training yields more significant performance improvements for smaller 3D fine-tuning datasets. In order to better showcase the effectiveness of pre-training, we use RayIoU as the evaluation metric, and the results obtained are as follows:

Table 12: **Detailed 3D occupancy prediction performance of different data scale utilized in pre-training and fine-tuning stage.** We use RayIoU as the evaluation metric (Liu et al., 2023).

| Fine-tuning | Pre-training | RayIoU | Large Static RayIoU | | | |
| --- | --- | --- | --- | --- | --- | --- |
| | | | Overall | 1m | 2m | 4m |
| 150 Scenes | × | 29.5 | 32.5 | 26.2 | 32.9 | 38.5 |
| | 700 Scenes | 33.4 (+3.9) | 36.6 (+4.1) | 30.2 | 37.0 | 42.5 |
| 450 Scenes | × | 35.1 | 37.9 | 31.5 | 38.3 | 44.0 |
| | 700 Scenes | 37.8 (+2.7) | 40.2 (+2.3) | 33.1 | 40.6 | 46.8 |
| 700 Scenes | × | 36.8 | 39.5 | 32.4 | 39.9 | 46.2 |
| | 700 Scenes | 39.0 (+2.2) | 41.4 (+1.9) | 34.7 | 41.9 | 47.6 |

As shown in Table 12, when using RayIoU as the evaluation metric, the improvements in overall RayIoU and RayIoU for large categories follow a similar trend, indicating that as the scale of the 3D fine-tuning dataset increases, the benefits of 2D pre-training do indeed gradually diminish.

A.3    SELF-SUPERVISED 4D OCCUPANCY FORECASTING AND MOTION PLANNING

Instead of generating occupancy predictions through the occupancy head $\mathcal{H}$, we support an alternative approach by utilizing the attribute projection head $\mathcal{P}$. Specially, by setting a threshold value $\tau$ for the 3D volume density field $\tilde{\sigma}$ of the scene, we can determinate whether a voxel is occupied. Subsequently, the semantic occupancy of the voxel $v_k$ can be formulated as:

$$\hat{Y}(v_k) = \text{argmax}(\tilde{s}(v_k)), \ \text{if } \tilde{\sigma}(v_k) \geq \tau, \tag{10}$$

where $\tilde{s}$ denotes the semantic field of the scene, and we regard $v_k$ as non-occupied if $\tilde{\sigma}(v_k) < \tau$.

In this manner, we can also obtain occupancy predictions during the pre-training stage. In other words, PreWorld is capable of engaging in self-supervised tasks as well. Therefore, to validate its performance as a self-supervised 3D occupancy world model, we compare it against state-of-the-art self-supervised methods on both 4D occupancy forecasting and motion planning tasks on the Occ3D-nuScenes dataset (Tian et al., 2024), denoting as **PreWorld-S**.

**4D Occupancy Forecasting.**    Table 13 presents the 4D occupancy forecasting performance of PreWorld-S compared to previous self-supervised approach of OccWorld (Zheng et al., 2023). In comparison to OccWorld-S, our approach yields significant outcomes. The IoU over the future 3-second interval nearly doubles, while the average future mIoU demonstrates an remarkable increase of over 1300%, soaring from 0.26 to 3.78. These results highlight the superiority of our method

in self-supervised learning and open up more possibilities for future research on the architecture of self-supervised 3D occupancy world models.

Table 13: **Self-supervised 4D occupancy forecasting performance on the Occ3D-nuScenes dataset.** We take the self-supervised vision-centric approach of OccWorld (Zheng et al., 2023) as baseline for fair comparison. Aux. Sup. represents auxiliary supervision apart from the ego trajectory. Avg. reprersents the average performance of that in 1s, 2s, and 3s. The best and second-best performances are represented by **bold** and underline respectively.

| Method | Aux. Sup. | mIoU (%) ↑ | | | | IoU (%) ↑ | | | |
|---|---|---|---|---|---|---|---|---|---|
| | | 1s | 2s | 3s | Avg. | 1s | 2s | 3s | Avg. |
| OccWorld-S | None | 0.28 | 0.26 | 0.24 | 0.26 | 5.05 | 5.01 | 4.95 | 5.00 |
| **PreWorld-S (Ours)** | 2D Labels | **4.36** | **3.72** | **3.27** | **3.78** | **9.49** | **9.17** | **8.90** | **9.19** |

Table 14: **Self-supervised motion planning performance on the Occ3D-nuScenes dataset.** We take the self-supervised vision-centric approach of OccWorld (Zheng et al., 2023) as baseline for fair comparison. † represents training and inference with ego state information introduced. The bestperformances are represented by **bold**.

| Method | Aux. Sup. | L2(m) (%) ↓ | | | | Collision Rate (%) ↓ | | | |
|---|---|---|---|---|---|---|---|---|---|
| | | 1s | 2s | 3s | Avg. | 1s | 2s | 3s | Avg. |
| OccWorld-S† | None | 0.67 | 1.69 | 3.13 | 1.83 | **0.19** | 1.28 | 4.59 | 2.02 |
| **PreWorld-S (Ours)** | 2D Labels | 0.66 | 1.49 | 2.60 | 1.58 | 0.57 | 1.26 | 2.23 | 1.35 |
| **PreWorld-S (Ours)†** | 2D Labels | **0.20** | **0.61** | **1.57** | **0.79** | 0.57 | **0.64** | **1.42** | **0.88** |

**Motion Planning.** As illustrated in Table 14, PreWorld-S significantly surpasses OccWorld-S on both metrics even without the incorporation of ego-state information. When ego-state information is introduced (indicated in gray), the performance of our self-supervised approach has received a notable enhancement, yielding results comparable to or outperforming those fully-supervised methods such as OccWorld-D. These findings once again demonstrate the effectiveness of our approach.

## B    MORE VISUALIZATIONS

We provide additional visualized comparison in this section.

Fig 5 shows more qualitative results of 3D occupancy prediction task compared with the latest fully-supervised method SparseOcc (Liu et al., 2023) and self-supervised method RenderOcc (Pan et al., 2024), further substantiating the robustness of our PreWorld model and the effectiveness of our novel two-stage training paradigm. The red boxes highlight fine-grained details of the 3D occupancy predictions and the ground truth, while the orange boxes mark holistic structure of an area within the scene. Compared to prior approaches, PreWorld demonstrates superior performance in preserving the structural information of the scene and capturing fine-grained details. In contrast, RenderOcc struggles with comprehending the scene structure accurately and exhibits inaccurate predictions for unsupervised occluded regions. SparseOcc, on the other hand, fails to effectively predict small objects like *poles* and long-tailed objects like *construction vehicles*, resulting in detail loss. These findings are consistent with the observations in the main text.

In Fig 6, we further provide a detailed showcase of the prediction results for both visible and occluded regions. Consistent with the quantitative analysis in Section A.1, it can be observed that RenderOcc (Pan et al., 2024) tends to predict thicker surfaces for large static categories. However, while this approach may lead to higher mIoU scores, its predictions for occluded regions are chaotic, indicating a lack of true understanding of the scene structure. On the contrary, our PreWorld makes more cautious predictions for occluded regions, demonstrating a more comprehensive understanding of the holistic scene structure.

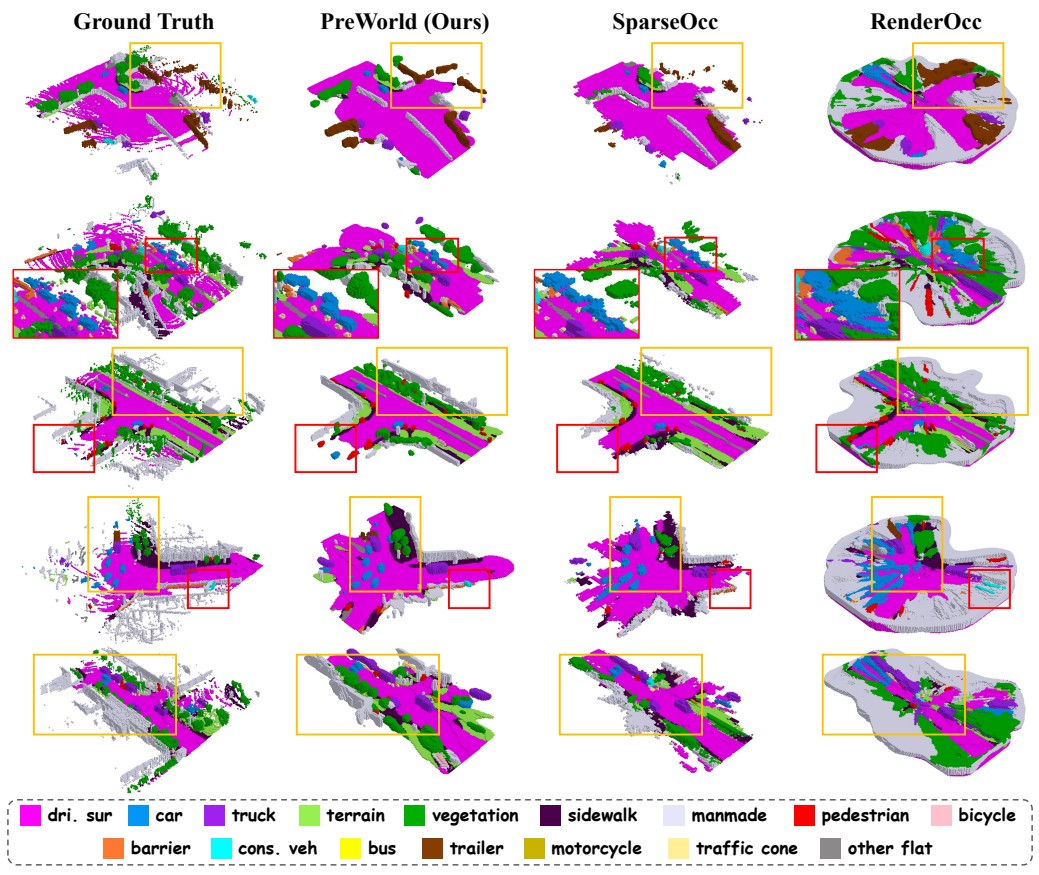

Figure 5: **More qualitative results of 3D occupancy prediction on the Occ3D-nuScenes validation set.** The **holistic structure** and **fine-grained details** of the scene are highlighted by orange boxes and red boxes respectively.

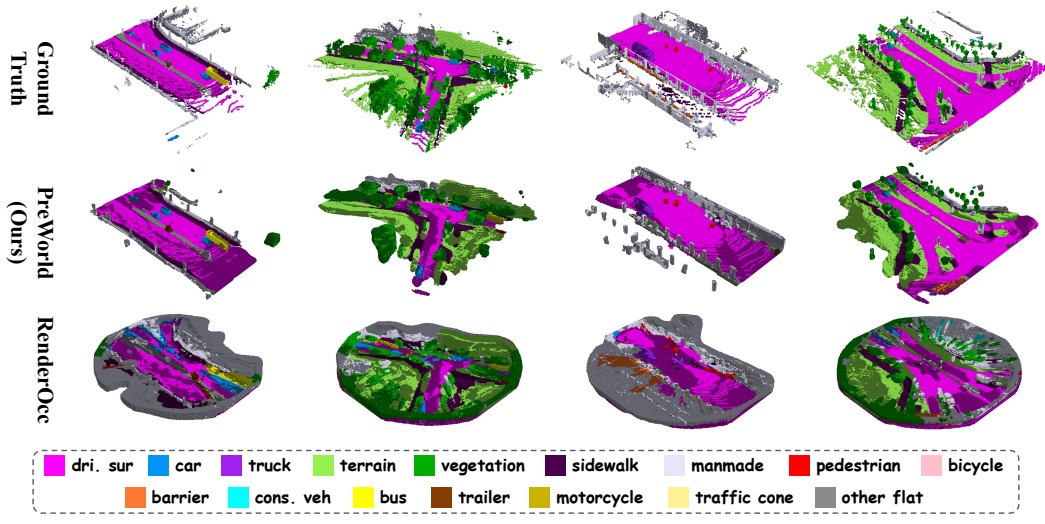

Figure 6: **More qualitative results of 3D occupancy prediction on the Occ3D-nuScenes validation set.** The shaded area represents occluded regions where the voxels are not included in the evaluation. In contrast to RenderOcc, our PreWorld makes more cautious predictions for occluded regions, tending to preserve the overall structure of the scene.

