# OpenReview forum: "Semi-Supervised Vision-Centric 3D Occupancy World Model for Autonomous Driving"
_ICLR.cc/2025/Conference — ICLR 2025 Poster_

### Official Review · Reviewer_BYPK · 2024-10-31

**Soundness:** 3
**Presentation:** 3
**Contribution:** 2
**Rating:** 8
**Confidence:** 4

**Summary:**

The paper proposes PreWorld, a semi-supervised, vision-centric 3D occupancy model that leverages 2D labels through a novel two-stage training approach. In the self-supervised pre-training stage, an attribute projection head is used to enable temporal supervision from 2D labels, while the fully-supervised fine-tuning stage introduces a state-conditioned forecasting module to directly predict future occupancy and ego trajectory.

**Strengths:**

The paper presents a comprehensive approach, from the overall architectural design to thorough experimental validation:

a). The paper is well-written and easy to follow, presenting a method that is both well-motivated and clearly explained.

b). The proposed state-conditioned forecasting module is interesting.

**Weaknesses:**

a). Although PreWorld leverages 2D labels to reduce the cost of 3D annotations, the paper only utilizes the nuScenes dataset for training, which does not fully demonstrate the method’s advantages. The semi-supervised approach should be highlighted by showcasing its benefits in terms of data efficiency.

b). Different tasks in occupancy predictions (4D occupancy forecasting, motion planning...) may have interdependent effects, where optimizing one module could potentially negatively impact others. The paper should provide a clear discussion on how to achieve a balance among these tasks and address potential interactions between the modules.

c). PreWorld’s two-stage training process may increase model complexity. This could lead to higher computational demands, potentially affecting the feasibility of real-time application in autonomous driving. The paper should further validate the feasibility of this method in terms of real-time inference performance.

**Questions:**

See weaknesses.

---

> ### Author Response · Authors · 2024-11-20
>
> We thank Reviewer BYPK for their insightful and positive review.
>
> > Although PreWorld leverages 2D labels to reduce the cost of 3D annotations, the paper only utilizes the nuScenes dataset for training, which does not fully demonstrate the method’s advantages. The semi-supervised approach should be highlighted by showcasing its benefits in terms of data efficiency.
>
> Thank you for this valuable suggestion. Due to the limited scale of the nuScenes dataset, we validate the effectiveness and scalability of our method by reducing the scale of 3D occupancy labels for fine-tuning in Table 5. The results in the table indicate that through 2D pre-training, the model utilizing a smaller scale of 3D annotated data (450 Scenes) can achieve comparable performance to the one with the whole dataset (700 Scenes).
>
> To better showcase the advantages of our method, we conduct further experiments on the nuPlan dataset, utilizing 3D occupancy annotations from the OpenScene benchmark. Specifically, we randomly select 750 and 150 scenes for training and evaluation. For pre-training, we randomly select supplemental scenes to increase the total pre-training data scale to 1,500 and 1,5000 scenes. **It is noteworthy that we only employ the images from these scenes.** The results are shown as follows:
>
> | Method                            | IoU               | mIoU              |
> | --------------------------------- | ----------------- | ----------------- |
> | OccNet-T [1]                      | 32.82             | 11.52             |
> | OccNet-B [1]                      | 34.22             | 13.60             |
> | **PreWorld (Ours)**               | 38.25             | 18.63             |
> | **+ Pre-training (750 scenes)**   | 38.40 (+0.15)     | 19.06 (+0.43)     |
> | **+ Pre-training (1500 scenes)**  | 38.55 (+0.30)     | 19.29 (+0.66)     |
> | **+ Pre-training (15000 scenes)** | **38.62** (+0.37) | **19.85** (+1.22) |
>
> These results demonstrate that at a larger scale of pre-training data, our approach can lead to more significant performance enhancements, thus providing stronger evidence for the effectiveness of pre-training. Considering that the cost of 2D image data collection is significantly lower than 3D occupancy data annotation, the advantages of our method are quite remarkable.
>
> [1] Tong, et al. "Scene as occupancy." ICCV 2023.

---

> > ### Author Response · Authors · 2024-11-20
> >
> > > Different tasks in occupancy predictions (4D occupancy forecasting, motion planning...) may have interdependent effects, where optimizing one module could potentially negatively impact others. The paper should provide a clear discussion on how to achieve a balance among these tasks and address potential interactions between the modules.
> >
> > This is an interesting perspective, and thank you for bringing that to our attention. We conduct further experiments by jointly optimizing the 4D occupancy forecasting and motion planning tasks to validate the gains of joint training for individual tasks.
> >
> > | Ego  | SSP  | TS   | mIoU  |      |      |                  | IoU   |       |       |                   |
> > | ---- | ---- | ---- | ----- | ---- | ---- | ---------------- | ----- | ----- | ----- | ----------------- |
> > | 1s   | 2s   | 3s   | Avg.  | 1s   | 2s   | 3s               | Avg.  |       |       |                   |
> > |      |      |      | 11.12 | 7.73 | 5.89 | 8.25             | 22.91 | 20.31 | 17.84 | 20.35             |
> > | √    |      |      | 11.17 | 8.54 | 6.83 | 8.85 (+0.60)     | 23.27 | 20.83 | 18.51 | 20.87 (+0.52)     |
> > | √    |      | √    | 11.69 | 8.72 | 6.77 | 9.06 (+0.81)     | 23.01 | 20.79 | 18.84 | 20.88 (+0.53)     |
> > | √    | √    |      | 11.58 | 9.14 | 7.34 | 9.35 (+1.10)     | 23.27 | 21.41 | 19.49 | 21.39 (+1.04)     |
> > | √    | √    | √    | 12.27 | 9.24 | 7.15 | **9.55** (+1.30) | 23.62 | 21.62 | 19.63 | **21.62** (+1.27) |
> >
> > Here TS denotes trajetory supervision. The results in the table demonstrate that joint training can further enhance the performance of the model on the 4D occupancy prediction task, with and without pre-training. The incorporation of motion planning task boosts the previous best results from the average mIoU of 8.85 and 9.35 to 9.06 and 9.55, respectively.
> >
> > | Input                | Sup.                         | L2       |          |          |          | Collision Rate |          |          |          |
> > | -------------------- | ---------------------------- | -------- | -------- | -------- | -------- | -------------- | -------- | -------- | -------- |
> > |                      |                              | 1s       | 2s       | 3s       | Avg.     | 1s             | 2s       | 3s       | Avg.     |
> > | 2D Image & Ego state | Traj GT                      | **0.20** | 0.34     | 0.80     | 0.45     | 0.50           | **0.62** | 0.90     | 0.67     |
> > | 2D Image & Ego state | Traj GT & 2D Labels & 3D Occ | 0.22     | **0.30** | **0.40** | **0.31** | **0.21**       | 0.66     | **0.71** | **0.53** |
> >
> > The results in the table demonstrate that by supervising the occupancy forecasting results simultaneously, the motion planning capabilities of our model are further enhanced in both L2 performance (0.45->0.31) and collision rates (0.67->0.53).
> >
> > In summary, our further experiments indicate that jointly optimizing the 4D occupancy forecasting and motion planning tasks leads to additional improvements in model performance compared to optimizing them separately. We will include the results and analysis of this part in our revised manuscript.
> >
> > > PreWorld’s two-stage training process may increase model complexity. This could lead to higher computational demands, potentially affecting the feasibility of real-time application in autonomous driving. The paper should further validate the feasibility of this method in terms of real-time inference performance.
> >
> > Thank you for pointing this out. The attribute projection head and volume rendering module used in the pre-training stage are not involved in computations during the fine-tuning and inference stages. Therefore, they can be considered as **plug-and-play** modules that do not negatively impact the computational efficiency of the base model.

---

> > > ### Comment · Reviewer_BYPK · 2024-11-26
> > >
> > > I appreciate the authors' efforts in clarifying the issues in the rebuttal.  I would like to raise the score to 8.

---

### Official Review · Reviewer_fN92 · 2024-11-01

**Soundness:** 2
**Presentation:** 3
**Contribution:** 2
**Rating:** 6
**Confidence:** 4

**Summary:**

The paper propose a semi-supervised method called Pre-World for occupancy forecasting problem. Pre-world first train the occupancy network in a self-supervised manner with the pre-training goal to render future image observation. Then Pre-world train the occupancy network with temporal occupancy labels to further improve the performance. Experiment results on NuScenes dataset show that the proposed method to some extend improve the performance compared to training from scratch.

**Strengths:**

The paper is easy to follow. The proposed Preworld is able to utilize unlabeled data for pre-training and improve the performance of the downstream tasks. The experiments are extensive.

**Weaknesses:**

1) The performance improvement seems not that significant. Generally, there are two kinds of improvement that pre-training can bring. The first one is to accelerate convergence in the downstream task. The second one is to improve sample efficiency in the downstream task, that is to improve the performance at convergence. The reviewer are concerned about whether the improvement presented in the experiment part is related to the acceleration of convergence.

2) It can be found in Table 1 that after pre-training, ious of some categories like driving surface and manmade drop. Also, compared to previous method, performance on some categories including manmade and vegetation are lower. But there is no analysis about these.

**Questions:**

1)  Is it possible for the authors to provide experiments on training longer for both randomly initialization and pre-trained ones until they both converge in the downstream task?

2) Could the authors provide analysis on the detailed category performance?

---

> ### Author Response · Authors · 2024-11-20
>
> We thank Reviewer fN92 for their thorough and insightful review.
>
> > The performance improvement seems not that significant. Generally, there are two kinds of improvement that pre-training can bring. The first one is to accelerate convergence in the downstream task. The second one is to improve sample efficiency in the downstream task, that is to improve the performance at convergence. The reviewer are concerned about whether the improvement presented in the experiment part is related to the acceleration of convergence.
> >
> > Is it possible for the authors to provide experiments on training longer for both randomly initialization and pre-trained ones until they both converge in the downstream task?
>
> This is an interesting perspective, and thank you for bringing that to our attention. Due to time constraints, when conducting experiments on smaller datasets, models fine-tuned on 150 scenes and 450 scenes are trained for 24 and 18 epochs, respectively, while the model on the full dataset is trained for 12 epochs. Considering the ratio of data reduction to extended training time, we believe that we do not allocate sufficient additional training time for experiments on the smaller datasets. The results after training longer are shown as follows (using mIoU as the evaluation metric):
>
> | Fine-tuning | Pre-training | 12        | 18        | 24        | 36        | 48        | 60    |
> | ----------- | ------------ | --------- | --------- | --------- | --------- | --------- | ----- |
> | 150 scenes  | ×            | 11.18     | 13.85     | 18.66     | 29.30     | **30.26** | 30.00 |
> |             | 700 scenes   | 13.01     | 21.83     | 25.02     | **31.65** | 31.56     | 30.98 |
> | 450 scenes  | ×            | 25.54     | 31.99     | 32.89     | **33.32** |           |       |
> |             | 700 scenes   | 29.52     | 33.37     | **34.19** | 34.08     |           |       |
> | 700 scenes  | ×            | 33.95     | **33.99** |           |           |           |       |
> |             | 450 scenes   | **34.28** | 34.15     |           |           |           |       |
> |             | 700 scenes   | 34.69     | **34.89** |           |           |           |       |
>
> As shown in the results, we believe that **pre-training has benefited the model in both aspects**. Taking models fine-tuned on 150 scenes as an example, it can be observed that during the first 24 epochs, employing pre-training accelerates the convergence of the model. Subsequently, both models have converged, with the pre-trained model still maintaining an advantage in prediction performance. We will include these results and analysis in our revised manuscript, and thank you for pointing this out.

---

> > ### Author Response · Authors · 2024-11-20
> >
> > > It can be found in Table 1 that after pre-training, ious of some categories like driving surface and manmade drop. Also, compared to previous method, performance on some categories including manmade and vegetation are lower. But there is no analysis about these.
> > >
> > > Could the authors provide analysis on the detailed category performance?
> >
> > Thank you for pointing this out. The results presented in our table demonstrate the **overall enhancement** of prediction performance from pre-training, benefiting both **small objects** (such as *motorcycles* and *pedestrians*) and **large dynamic objects** (such as *buses* and *construction vehicles*). We believe this is due to the extensive supervision from 2D data during the pre-training stage.
> >
> > Conversely, **performance declines** are mainly concentrated within **large static categories** (such as *driveable surface* and *manmade*). We believe that the underlying reason for this situation is that common practices use visible masks during evaluation, calculating mIoU only for voxels within the visible region. As a result, existing models tend to predict thicker surfaces for large static categories. **Since the current metric does not penalize predictions outside the visible region, models often achieve higher scores through this strategy.** However, this can result in the generation of unreasonable scene structures in predictions.
> >
> > To address this, SparseOcc [1] introduces **the RayIoU metric**, which involves emulating LiDAR rays and projecting them onto 3D occupancy predictions and ground truth to obtain semantic labels for intersected voxels and the traveled distance of the rays as depth information. The semantic labels are utilized to calculate the IoU between predictions and ground truth for each semantic class, with multiple thresholds (1m, 2m, 4m) set to constrain the difference in depth between predictions and ground truth.
> >
> > **The benefit of using RayIoU is that the model can no longer gain an advantage by predicting a thicker surface.** Therefore, we believe it is a more reasonable evaluation metric for large static categories. We report the detailed performance on large static categories in the table below:
> >
> > | Method         | GT    | RayIoU   | barrier  | driveable surface | other flat | sidewalk | terrain  | manmade  | vegetation |
> > | -------------- | ----- | -------- | -------- | ----------------- | ---------- | -------- | -------- | -------- | ---------- |
> > | RayIoU_1m      |       |          |          |                   |            |          |          |          |            |
> > | PreWorld       | 3D    | 30.0     | 39.4     | 56.4              | 24.2       | 25.5     | 23.8     | 32.9     | 23.2       |
> > | + Pre-training | 2D+3D | **32.5** | **40.1** | **57.8**          | **29.8**   | **27.2** | **27.3** | **35.1** | **25.9**   |
> > | RayIoU_2m      |       |          |          |                   |            |          |          |          |            |
> > | PreWorld       | 3D    | 37.2     | 44.6     | 64.4              | 29.2       | 30.7     | 31.3     | 43.4     | 36.0       |
> > | + Pre-training | 2D+3D | **39.6** | **45.4** | **65.9**          | **34.3**   | **32.7** | **34.6** | **44.7** | **37.8**   |
> > | RayIoU_4m      |       |          |          |                   |            |          |          |          |            |
> > | PreWorld       | 3D    | 41.9     | 46.6     | 72.4              | 33.2       | 35.6     | 38.1     | 49.7     | **47.3**   |
> > | + Pre-training | 2D+3D | **44.0** | **47.4** | **74.3**          | **38.3**   | **37.7** | **41.1** | **50.5** | 47.2       |
> >
> > As shown in the table, **the pre-trained model surpasses the model without pre-training on almost all large static categories across all thresholds.** The results under the RayIoU metric indicate that pre-training steers the model towards predicting more plausible scene structures. Therefore, overall, we believe that pre-training has not compromised model performance but has yielded gains across various aspects. In summary,
> >
> > * For **small objects** and **large dynamic objects**, pre-training can lead to significant performance boost by introducing extensive supervision from 2D data.
> > * For **large static categories**, pre-training can encourage model to generate more reasonable predictions, which can be reflected in the RayIoU metric.

---

> > > ### Author Response · Authors · 2024-11-20
> > >
> > > Furthermore, the model performance on the RayIoU metric also helps us understand why our model may not outperform baseline models in some categories. Our method do not achieve the SOTA performance in five categories, **all of which are concentrated in the large static categories**. The best performance in three of the categories is achieved by RenderOcc [2], which, as mentioned earlier, **has a tendency to predict thicker surfaces for better mIoU scores**. Here, we provide a visualization result comparing our model with RenderOcc and SparseOcc: https://www.hostize.com/v/Ta5ND-IuCV
> > >
> > > As shown in the figure, **RenderOcc can not generate reasonable scene structure even with a high mIoU score**. The remaining two SOTA results are achieved by OccFlowNet [3]. Given that OccFlowNet's model framework is very similar to RenderOcc, we can reasonably speculate that OccFlowNet also faces similar issues. Additionally, OccFlowNet is not part of publicly published works in academic journals or conferences.
> > >
> > > Therefore, we have surpassed all published baseline methods in almost all categories and explained why we may not perform as well as RenderOcc on this metric, for our PreWorld tends to generate a more reasonable scene structure.
> > >
> > > [1] Liu, et al. "Fully sparse 3d panoptic occupancy prediction." ECCV 2024.
> > >
> > > [2] Pan, et al. "Renderocc: Vision-centric 3d occupancy prediction with 2d rendering supervision." ICRA 2024.
> > >
> > > [3] Boeder, Simon, Fabian Gigengack, and Benjamin Risse. "Occflownet: Towards self-supervised occupancy estimation via differentiable rendering and occupancy flow."

---

> > ### Comment · Reviewer_fN92 · 2024-11-21
> > **Thank you for the additional results and there is a further question.**
> >
> > The reviewer appreciate the authors' effort on additional experiments. The results show that pre-training actually improve sample efficiency. But one remaining question is pre-training on 700 scenes and downstream to 700 scenes brings more (or comparable) improvement (0.9 mIoU) than downstream to 450 scenes (0.87 mIoU), which is a bit confusing because generally as the ratio of pre-training data to fine-tuning data increases, the performance gain also increases. Is there any analysis on this?

---

> > > ### Author Response · Authors · 2024-11-23
> > >
> > > > But one remaining question is pre-training on 700 scenes and downstream to 700 scenes brings more (or comparable) improvement (0.9 mIoU) than downstream to 450 scenes (0.87 mIoU), which is a bit confusing because generally as the ratio of pre-training data to fine-tuning data increases, the performance gain also increases. Is there any analysis on this?
> > >
> > > Thank you for bringing this up. We believe the reason for this still lies in the fact that, **for large categories, existing evaluation metric does not adequately reflect the model's actual performance.** Therefore, we have detailed the corresponding mIoU for large categories and small objects as follows:
> > >
> > > | Fine-tuning | Pre-training | mIoU          | Large Categories mIoU | Small Objects mIoU |
> > > | ----------- | ------------ | ------------- | --------------------- | ------------------ |
> > > | 150 scenes  | ×            | 30.26         | 35.82                 | 24.08              |
> > > |             | 700 scenes   | 31.65 (+1.39) | 37.18 (+1.36)         | 25.50 (+1.42)      |
> > > | 450 scenes  | ×            | 33.32         | 38.40                 | 26.29              |
> > > |             | 700 scenes   | 34.19 (+0.87) | 39.04 (+0.64)         | 27.44 (+1.15)      |
> > > | 700 scenes  | ×            | 33.99         | 39.83                 | 26.98              |
> > > |             | 700 scenes   | 34.89 (+0.90) | 40.72 (+0.89)         | 27.96 (+0.98)      |
> > >
> > > It can be observed that the mIoU for large categories does not always effectively reflect the performance improvement of the model. For the model fine-tuned with 450 scenes, pre-training leads to a 0.64 increase in mIoU for large categories, while the model fine-tuned with 700 scenes sees an increase of 0.89. In contrast, the increase in mIoU for small objects can better reflect the effectiveness of pre-training, aligning with the expectations: **2D pre-training yields more significant performance improvements for smaller 3D fine-tuning datasets.**
> > >
> > > In order to better showcase the effectiveness of pre-training, we use RayIoU as the evaluation metric, and the results obtained are as follows:
> > >
> > > | Fine-tuning | Pre-training | RayIoU      | Large RayIoU | Large RayIoU_1m | Large RayIoU_2m | Large RayIoU_4m |
> > > | ----------- | ------------ | ----------- | ----------------------- | --------------- | --------------- | --------------- |
> > > | 150 scenes  | ×            | 29.5        | 32.5                    | 26.2            | 32.9            | 38.5            |
> > > |             | 700 scenes   | 33.4 (+3.9) | 36.6 (+4.1)             | 30.2            | 37.0            | 42.5            |
> > > | 450 scenes  | ×            | 35.1        | 37.9                    | 31.5            | 38.3            | 44.0            |
> > > |             | 700 scenes   | 37.8 (+2.7) | 40.2 (+2.3)             | 33.1            | 40.6            | 46.8            |
> > > | 700 scenes  | ×            | 36.8        | 39.5                    | 32.4            | 39.9            | 46.2            |
> > > |             | 700 scenes   | 39.0 (+2.2) | 41.4 (+1.9)             | 34.7            | 41.9            | 47.6            |
> > >
> > > When using RayIoU as the evaluation metric, the improvements in overall RayIoU and RayIoU for large categories follow a similar trend, indicating that as the scale of the 3D fine-tuning dataset increases, the benefits of 2D pre-training do indeed gradually diminish.

---

> > > > ### Comment · Reviewer_fN92 · 2024-11-25
> > > > **Thank you for the explanation. The authors have addressed most of the reviewer's concern.**
> > > >
> > > > Thank you for the explanation! The reviewer believe that the updated results and explanation can be further added to the revision. Most of the concerns are addressed and the reviewer want to raise the score to 6.

---

### Official Review · Reviewer_GcMx · 2024-11-02

**Soundness:** 2
**Presentation:** 2
**Contribution:** 2
**Rating:** 6
**Confidence:** 4

**Summary:**

The paper presents "PreWorld," a semi-supervised model aimed at improving 3D occupancy prediction and 4D forecasting for autonomous driving. It leverages both 2D and 3D data in a two-stage training approach, including a self-supervised pre-training phase using 2D labels and a fully-supervised fine-tuning phase with 3D labels. The authors introduce a state-conditioned forecasting module that directly predicts future occupancy and vehicle trajectory. Tests on the Occ3D-nuScenes dataset show PreWorld achieves competitive results.

**Strengths:**

The paper provides a thorough evaluation of different methods on the Occ3D-nuScenes dataset. The experiments shows that the proposed model and training method can achieve competitive results across 3D occupancy prediction, 4D forecasting, and motion planning tasks.

**Weaknesses:**

1. The contributions are largely incremental. While the state-conditioned forecasting module (Section 3.2) enables volume feature preservation for rendering, this approach aligns closely with existing practices in similar tasks​ [1][2]. The self-supervision in Section 3.3 extends volume rendering to 4D but follows typical methodologies in related work​ [3].
2. The figures are difficult to interpret due to insufficient captions. For instance, Figures 2 and 3 lack details, making it unclear how the components operate or interact. The difference between solid and dashed arrows in Figures 1, 2, and 3 is not explained. Figure 2 leaves confusion about whether the two stages are jointly or sequentially trained.
3. The authors claim 2D rendering as self-supervision, but the best result comes from utilizing depth and semantic labels, which is not self-supervision, and invalidates "semi-supervision" in the title.
4. The performance improvement is minimal. For example, the mIoU improvment for 3d occupancy prediction is only about 2% compared to OccFlowNet. And there is no significant improvement in other experiments and metrics

[1] Khurana, et al. "Point cloud forecasting as a proxy for 4d occupancy forecasting." CVPR 2023.

[2] Liu, et al. "Lidar-based 4d occupancy completion and forecasting." IROS 2024.

[3] Yang, et al. "Emernerf: Emergent spatial-temporal scene decomposition via self-supervision."

**Questions:**

The proposed two stages can be trained jointly, why train them in two stages? Can the authors provide any motivation on this?

---

> ### Author Response · Authors · 2024-11-20
>
> We thank Reviewer GcMx for their thorough and insightful review.
>
> > The contributions are largely incremental. While the state-conditioned forecasting module (Section 3.2) enables volume feature preservation for rendering, this approach aligns closely with existing practices in similar tasks [1] [2]. The self-supervision in Section 3.3 extends volume rendering to 4D but follows typical methodologies in related work [3].
>
> [1] [2] focus on the LiDAR-based 4D occupancy forecasting task, which involve **different input modalities** and **different processing pipelines** compared to the vision-centric 3D occupancy prediction task (like RenderOcc [4] and SparseOcc [5]) and the vision-centric 4D occupancy forecasting task (like OccWorld and OccLLaMA) that we are concerned with. For example, **the method in [1] does not employ an autoregressive approach to predict future scenes**, whereas our work follows the practices of OccWorld and OccLLaMA, adopting an autoregressive manner.
>
> Furthermore, it is important to note that **the forecasting module itself is not our novel contribution**. As highlighted in Section 3.1, the similar design is also present in OccWorld and OccLLaMA. The key distinction lies in their approach of **obtaining future occupancy from current occupancy**, whereas our approach involves **obtaining future volume features from current volume features**. Therefore, **our contribution is providing a more end-to-end framework for vision-centric 4D occupancy forecasting tasks**. Through a simple forecasting module composed of just several MLPs, we have achieved SOTA performance, demonstrating the effectiveness of this framework.
>
> In summary, we do not think that these two articles are very similar to ours. We will consider including these two articles (or more) to the related work section as a supplement to the LiDAR-based 4D occupancy forecasting task.
>
> Moreover, we provide several reasons why our work differs from EmerNeRF [3]:
>
> a) **Different usage of the attribute projection head**. EmerNeRF employs the projection head to directly predict the RGB and occupancy of future scenes, whereas we utilize the projection head simply for model supervision during pre-training. The future occupancy is generated by a dedicated occupancy head. This difference is related to our motivation: we aim to decouple the use of 2D and 3D data. The projection head provides supervision during pre-training using large-scale 2D data, while the occupancy head can be supervised using a smaller scale of 3D data.
>
> b) **Different methods for predicting future scenes**. EmerNeRF utilizes a dynamic head and a flow head to learn scene dynamics. In contrast, we still employ a dedicated future forecasting module to achieve this without adding additional training burden to the projection head.
>
> c) **Different design approaches**. EmerNeRF employs over eight MLPs to represent various aspects of the scene such as static features, dynamic features, colors, sky colors, shadows, etc. This design complexity is too intricate for our purposes. In contrast, PreWorld uses only three MLPs to represent the color, semantics, and density of the scene. Since the projection head in our design is solely responsible for pre-training, the simplicity in design can be compensated by a sufficiently large data scale.
>
> Based on the above points, we believe that our work differs significantly from EmerNeRF as well. We will consider including this paper in the related work section as a supplementary reference for the vision-centric 4D occupancy forecasting task.
>
> [1] Khurana, et al. "Point cloud forecasting as a proxy for 4d occupancy forecasting." CVPR 2023.
>
> [2] Liu, et al. "Lidar-based 4d occupancy completion and forecasting." IROS 2024.
>
> [3] Yang, et al. "Emernerf: Emergent spatial-temporal scene decomposition via self-supervision."
>
> [4] Pan, et al. "Renderocc: Vision-centric 3d occupancy prediction with 2d rendering supervision." ICRA 2024.
>
> [5] Liu, et al. "Fully sparse 3d panoptic occupancy prediction." ECCV 2024.
>
> [6] Zheng, et al. "Occworld: Learning a 3d occupancy world model for autonomous driving." ECCV 2025.
>
> [7] Wei, et al. "Occllama: An occupancy-language-action generative world model for autonomous driving."

---

> > ### Author Response · Authors · 2024-11-20
> >
> > > The figures are difficult to interpret due to insufficient captions. For instance, Figures 2 and 3 lack details, making it unclear how the components operate or interact. The difference between solid and dashed arrows in Figures 1, 2, and 3 is not explained. Figure 2 leaves confusion about whether the two stages are jointly or sequentially trained.
> >
> > Thank you for bringing that to our attention, we will include more detailed captions in our revised manuscript.
> >
> > > The authors claim 2D rendering as self-supervision, but the best result comes from utilizing depth and semantic labels, which is not self-supervision, and invalidates "semi-supervision" in the title.
> >
> > **Our motivation is to leverage abundant and easily accessible 2D image sequences for large-scale self-supervised pre-training of the occupancy models.** However, due to the limited scale of the nuScenes dataset, the advantages of our approach are not prominently highlighted. Therefore, on the nuScenes dataset, we utilized additional 2D labels to better showcase the potential of pre-training. Furthermore, we believe that **annotating semantic and depth information on 2D images is still a cost-effective task compared to annotating 3D occupancy labels**. Moreover, **high-quality pseudo-labels generated by existing vision language models can also serve as substitutes**. As a result, we refer to this training paradigm that uses "large-scale 2D data" and "smaller scale 3D occupancy data" as "semi-supervision."
> >
> > We preliminarily validate the effectiveness and scalability of this paradigm by reducing the scale of 3D occupancy data during training, as shown in Table 5. For better illustration, we conduct further experiments on the nuPlan dataset, utilizing 3D occupancy annotations from the OpenScene benchmark. Specifically, we randomly select 750 and 150 scenes for training and evaluation. For pre-training, we randomly select supplemental scenes to increase the total pre-training data scale to 1,500 and 1,5000 scenes. **It is noteworthy that we only employ the images from these scenes.** The results are shown as follows:
> >
> > | Method                            | IoU               | mIoU              |
> > | --------------------------------- | ----------------- | ----------------- |
> > | OccNet-T [8]                      | 32.82             | 11.52             |
> > | OccNet-B [8]                      | 34.22             | 13.60             |
> > | **PreWorld (Ours)**               | 38.25             | 18.63             |
> > | **+ Pre-training (750 scenes)**   | 38.40 (+0.15)     | 19.06 (+0.43)     |
> > | **+ Pre-training (1500 scenes)**  | 38.55 (+0.30)     | 19.29 (+0.66)     |
> > | **+ Pre-training (15000 scenes)** | **38.62** (+0.37) | **19.85** (+1.22) |
> >
> > These results demonstrate that at a larger scale of pre-training data, our approach can lead to more significant performance enhancements, thus providing stronger evidence for the effectiveness and scalability of 2D self-supervised pre-training.
> >
> > [8] Tong, et al. "Scene as occupancy." ICCV 2023.

---

> > > ### Author Response · Authors · 2024-11-20
> > >
> > > > The performance improvement is minimal. For example, the mIoU improvment for 3d occupancy prediction is only about 2% compared to OccFlowNet. And there is no significant improvement in other experiments and metrics
> > >
> > > It is important to note that OccFlowNet [9] is not part of publicly published works in academic journals or conferences. We included it in the comparison just to ensure a more comprehensive presentation of occupancy-related works. **In comparison with published works like SparseOcc, our model achieved around a 12% performance improvement.**
> > >
> > > On the other hand, we believe that the current mIoU metric does not effectively showcase the advantages of our method. Common practices use visible masks during evaluation, calculating mIoU only for voxels within the visible region. As a result, existing models tend to predict thicker surfaces for large static categories, such as *driveable surface* and *manmade*. **Since the current metric does not penalize predictions outside the visible region, models often achieve higher scores through this strategy.** However, this can result in the generation of unreasonable scene structures in predictions. Here we provide a visualization result comparing our model with RenderOcc and SparseOcc: https://www.hostize.com/v/Ta5ND-IuCV
> > >
> > > As shown in the figure, **RenderOcc can not generate reasonable scene structure even with a high mIoU score**. Given that OccFlowNet's model framework is very similar to RenderOcc, **we can reasonably speculate that OccFlowNet also faces similar issues**.
> > >
> > > To address this, SparseOcc introduces the RayIoU metric, which we believe to be a more reasonable evaluation metric. As shown in Table 7 in the supplementary material, when utilizing RayIoU as the evaluation metric, our approach sets new SOTA performance by 2.6 RayIoU, with the proposed pre-training stage boosting model performance from 36.4 to 38.7, reaffirming the effectiveness of our training paradigm.
> > >
> > > Furthermore, we believe that **our method has made significant advancements in other tasks as well**. For 4D occupancy forecasting, PreWorld surpasses existing 3D occupancy world model with a rather simple module design. For motion planning, PreWorld achieves comparable L2 performance to some planning models even without ego-state information. By incorporating ego-state information, the motion planning capabilities of our model are significantly enhanced, surpassing other existing 3D occupancy world models by a notable margin.
> > >
> > > [9] Boeder, Simon, Fabian Gigengack, and Benjamin Risse. "Occflownet: Towards self-supervised occupancy estimation via differentiable rendering and occupancy flow."
> > >
> > > > The proposed two stages can be trained jointly, why train them in two stages? Can the authors provide any motivation on this?
> > >
> > > Our ultimate goal is to achieve superior performance by **conducting large-scale self-supervised pre-training of occupancy models using abundant unlabeled 2D image data**. This approach aims to excel in performance while minimizing the cost of annotating 3D occupancy data. Due to the limitations of the nuScenes dataset, we are only able to preliminarily validate the effectiveness and scalability of our training paradigm through Table 5. However, as shown in the table above, our training paradigm can achieve even more significant performance improvements on larger-scale datasets.
> > >
> > > Based on this motivation, we believe it is necessary to divide the training process into two stages: one using 2D image data (self-supervised pre-training stage) and the other using 3D occupancy data (fully-supervised fine-tuning stage). **This approach ensures the decoupling of our requirements for these two types of training data, such as using data with imbalanced scales, which joint training cannot achieve.** Furthermore, we aim for the 2D data to be sourced from various datasets, different acquisition environments, and diverse camera settings to truly enhance the generalization of the pre-training process. Subsequently, only a small amount of 3D data would be needed for fine-tuning in specific scenarios, a feat that joint training also cannot accomplish.

---

> ### Comment · Reviewer_GcMx · 2024-11-22
>
> I appreciate the insightful explanation and clarification provided by the author. However, I believe there are still two major concerns not addressed.
> 1. **Model performance**.
> * The authors dismiss OccFlowNet on the basis that it is "not published," which is an inadequate justification. OccFlowNet has already received 8 citations up-to-date, demonstrating its relevance and impact within the research community. Ignoring such a recognized work undermines the thoroughness of the evaluation and fails to acknowledge existing contributions that are pertinent to the study.
>
> * The introduction of RayIoU as an evaluation metric lacks proper justification. The authors do not explain in the main text of the paper way RayIoU is superior to the widely accepted mIoU. The authors should provide clear and convincing rationale in the paper for adopting this metric.
>
> * In both Table 7 and the supplementary figure (https://www.hostize.com/v/Ta5ND-IuCV), the absence of results for OccFlowNet prevents a comprehensive comparison. Without including OccFlowNet’s performance metrics, it is difficult to assess how the proposed method stands against existing approaches, thereby weakening the paper’s claims of superiority or innovation.
>
> * To robustly evaluate predictions outside the visible region, as the aurhors mentioned, the authors should have included results from the SSCBench [1] dataset, which provides ground truth labels beyond visible areas. The omission of this evaluation leaves a critical gap in demonstrating the method’s effectiveness in handling those scenarios.
>
> [1] Li et al., "Sscbench: A large-scale 3d semantic scene completion benchmark for autonomous driving." 2023.
>
> 2. **Method presentation**.
> * The rebuttal fails to provide additional explanations or clarifications for the figures. This lack of detail makes it difficult for readers to fully understand the proposed method, assess its validity, or replicate the results.
> * Without comprehensive explanations accompanying the figures, the methodological contributions remain vague. This lack of clarity hinders the ability to evaluate the robustness and originality of the proposed approach, raising concerns about the paper’s overall rigor and contribution to the field.

---

> > ### Author Response · Authors · 2024-11-26
> >
> > > The authors dismiss OccFlowNet on the basis that it is "not published," which is an inadequate justification. OccFlowNet has already received 8 citations up-to-date, demonstrating its relevance and impact within the research community. Ignoring such a recognized work undermines the thoroughness of the evaluation and fails to acknowledge existing contributions that are pertinent to the study.
> >
> > First and foremost, it is essential to emphasize that the ICLR policy (https://iclr.cc/Conferences/2025/FAQ) states, "**Authors are encouraged to cite and discuss all relevant papers, but they may be excused for not knowing about papers not published in peer-reviewed conference proceedings or journals, which includes papers exclusively available on arXiv.** " In other words, for works like OccFlowNet **that are solely available on arXiv and have not undergone peer review**, we are entitled not to compare our performance with them. Moreover, our performance has indeed surpassed this work.
> >
> > However, we do proceed with further experiments. **Since OccFlowNet does not provide pre-trained model weights, we conduct a reproduction experiment, yielding an mIoU of 32.18, which is lower than the reported 33.86.** Based on this result, our PreWorld achieved an improvement of nearly 8% in the mIoU metric.
> >
> > | Method                    | mIoU      | others    | barrier   | bicycle   | bus       | car       | construction vehicle | motorcycle | pedestrain | traffic cone | trailer   | truck     | driveable surface | other flat | sidewalk  | terrain   | manmade   | vegetation |
> > | ------------------------- | --------- | --------- | --------- | --------- | --------- | --------- | -------------------- | ---------- | ---------- | ------------ | --------- | --------- | ----------------- | ---------- | --------- | --------- | --------- | ---------- |
> > | SparseOcc                 | 30.90     | -         | -         | -         | -         | -         | -                    | -          | -          | -            | -         | -         | -                 | -          | -         | -         | -         | -          |
> > | OccFlowNet (reproduction) | 32.18     | 8.01      | 38.83     | 25.06     | 41.37     | 43.22     | 21.65                | 28.43      | 23.20      | 25.14        | 27.91     | 33.61     | 60.67             | 34.80      | 39.35     | 34.87     | 32.18     | 28.78      |
> > | OccFlowNet (reported)     | 33.86     | 8.00      | 37.60     | 26.00     | 42.10     | 42.50     | 21.60                | 29.20      | 22.30      | 25.70        | 29.70     | 34.40     | **64.90**         | 37.20      | **44.30** | **43.20** | **34.30** | **32.50**  |
> > | PreWorld (Ours)           | **34.69** | **11.81** | **45.01** | **26.29** | **43.32** | **47.71** | **24.23**            | **31.29**  | **27.41**  | **27.68**    | **30.62** | **35.64** | 63.71             | **37.27**  | 41.20     | 37.54     | 29.36     | 29.70      |
> >
> > More importantly, we evaluate the performance of OccFlowNet using the RayIoU metric, and the results are as follows:
> >
> > | Method                    | RayIoU   | RayIoU_1m | RayIoU_2m | RayIoU_4m |
> > | ------------------------- | -------- | --------- | --------- | --------- |
> > | OccFlowNet (reproduction) | 32.6     | 25.6      | 33.3      | 38.8      |
> > | SparseOcc                 | 36.1     | 30.2      | 36.8      | 41.2      |
> > | PreWorld (Ours)           | **38.7** | **32.5**  | **39.6**  | **44.0**  |
> >
> > **It can be observed that OccFlowNet surpasses SparseOcc in terms of mIoU, but significantly underperforms in terms of RayIoU compared to the latter, let alone our model.** These results validate our earlier assessment that OccFlowNet, similar to RenderOcc, faces a issue where they achieve higher mIoU scores in large static categories by predicting thicker surfaces but struggle to comprehend scene structures accurately, resulting in lower RayIoU scores.
> >
> > In summary, these extensive experimental results further demonstrate the superiority of our approach.

---

> > > ### Author Response · Authors · 2024-11-26
> > >
> > > > The introduction of RayIoU as an evaluation metric lacks proper justification. The authors do not explain in the main text of the paper way RayIoU is superior to the widely accepted mIoU. The authors should provide clear and convincing rationale in the paper for adopting this metric.
> > >
> > > In our revised manuscript, we conduct a more detailed analysis of the RayIoU metric and include additional experimental results to complement our findings. Moreover, it is necessary to emphasize that the introduction of the RayIoU metric is intended to provide a clearer explanation for the performance decline of our model under mIoU and why our model lags behind baseline models in certain categories. Therefore, to align with common practices [2-5], we continue to utilize the mIoU metric in the main text while offering a detailed analysis of the results under the RayIoU metric in Section A.1.
> > >
> > > [2] Zhang et al., "Occformer: Dual-path transformer for vision-based 3d semantic occupancy prediction", ICCV 2023.
> > >
> > > [3] Tian et al., "Occ3d: A large-scale 3d occupancy prediction benchmark for autonomous driving", NeurIPS 2023.
> > >
> > > [4] Li et al., "Fb-occ: 3d occupancy prediction based on forward-backward view transformation", CVPR 2023.
> > >
> > > [5] Pan et al., "Renderocc: Vision-centric 3d occupancy prediction with 2d rendering supervision", ICRA 2024.
> > >
> > > > In both Table 7 and the supplementary figure (https://www.hostize.com/v/Ta5ND-IuCV), the absence of results for OccFlowNet prevents a comprehensive comparison. Without including OccFlowNet’s performance metrics, it is difficult to assess how the proposed method stands against existing approaches, thereby weakening the paper’s claims of superiority or innovation.
> > >
> > > In our revised manuscript, we have added the RayIoU performance of OccFlowNet to Table 7.
> > >
> > > > To robustly evaluate predictions outside the visible region, as the aurhors mentioned, the authors should have included results from the SSCBench [1] dataset, which provides ground truth labels beyond visible areas. The omission of this evaluation leaves a critical gap in demonstrating the method’s effectiveness in handling those scenarios.
> > >
> > > **We need to clarify that Occ3D also provides ground truth labels for occluded regions.** However, during evaluation, we follow common practices [3-9] by using visible masks to exclude these regions. Our analysis of occluded regions is simply to illustrate why RenderOcc and OccFlowNet can achieve higher mIoU scores in certain categories. To maintain consistency with common practices, we will not include experiments with the SSCBench [1] dataset, as it is irrelevant to our objectives and not a dataset widely used in previous works [3-11].
> > >
> > > To better demonstrate the model's prediction results for both visible and occluded regions, we provide visualized comparisons with RenderOcc: https://www.hostize.com/v/cnpCwid2c4
> > >
> > > Consistent with our analysis, RenderOcc tends to predict thicker surfaces for large static categories. However, while this approach may lead to higher mIoU scores, its predictions for occluded regions are chaotic, indicating a lack of true understanding of the scene structure. On the contrary, our PreWorld makes more cautious predictions for occluded regions, demonstrating a more comprehensive understanding of the holistic scene structure.
> > >
> > > [1] Li et al., "Sscbench: A large-scale 3d semantic scene completion benchmark for autonomous driving." 2023.
> > >
> > > [6] Tong et al. "Scene as occupancy." ICCV 2023.
> > >
> > > [7] Liu et al. "Fully sparse 3d panoptic occupancy prediction." ECCV 2024.
> > >
> > > [8] Huang et al., "Selfocc: Self-supervised vision-based 3d occupancy prediction", CVPR 2024.
> > >
> > > [9] Zhang et al., "Occnerf: Self-supervised multi-camera occupancy prediction with neural radiance fields".
> > >
> > > [10] Zheng et al. "Occworld: Learning a 3d occupancy world model for autonomous driving." ECCV 2025.
> > >
> > > [11] Wei et al. "Occllama: An occupancy-language-action generative world model for autonomous driving."
> > >
> > > > The rebuttal fails to provide additional explanations or clarifications for the figures. This lack of detail makes it difficult for readers to fully understand the proposed method, assess its validity, or replicate the results.
> > > >
> > > > Without comprehensive explanations accompanying the figures, the methodological contributions remain vague. This lack of clarity hinders the ability to evaluate the robustness and originality of the proposed approach, raising concerns about the paper’s overall rigor and contribution to the field.
> > >
> > > In our revised manuscript, we have provided more detailed captions. For example, in Figures 1, 2, and 3, we have made the meaning of the dashed arrows clearer either within the figures or in the captions.

---

> > > > ### Author Response · Authors · 2024-12-02
> > > >
> > > > We believe we have sufficiently addressed your questions and concerns. With only two days remaining for discussion, we are eager to receive your response.

---

> > > > > ### Comment · Reviewer_GcMx · 2024-12-02
> > > > >
> > > > > I believe my concerns and questions are well addressed in the discussion.
> > > > > I have updated my rating.

---

> > > > > > ### Author Response · Authors · 2024-12-03
> > > > > >
> > > > > > Thank you for your feedback. However, we notice that despite addressing all your concerns, your rating remains negative. Please let us know about any parts of our paper that are still unclear to you or where you believe may need further enhancement. Again, we greatly appreciate your suggestions for improving our revised manuscript.

---

### Official Review · Reviewer_z3jn · 2024-11-05

**Soundness:** 3
**Presentation:** 3
**Contribution:** 2
**Rating:** 6
**Confidence:** 2

**Summary:**

This paper presents a semi-supervised vision-centric 3D occupancy world mdoel, PreWorld. It leverages 2D labels through two-stage training, self-supervised pretraining and supervised finetuning. In the pre-training stage, the method utilizes an attribute projection head to enable temporal supervision from 2D labels. The method also includes a state-conditioned forecasting module.

**Strengths:**

1. The self-supervised pre-training is simple and intuitive. The method projects predicted volume using attribute projection head, then use future 2D frames as supervision signal. It is effective by upgrading training strategy without introducting extra modules for inference.
2. The experiments are extensive to show superior performance in occupancy prediction and forecasting tasks.
3. The paper writing is clear, it is easy to read and follow.

**Weaknesses:**

1. The improvement with and with-out pre-training seems not very significant. The module sometimes hurt performance (In tab.1).
2. The ego state-conditioned module lacks of sufficient analysis and comparison.

**Questions:**

1. "Our Copy&Paste"  in Tab.6 is confusing for me. What is the setting of row 2 in Tab. 6 and what is difference between row 2 and copy & paste?
2. How to correctly interrept the result in Tab.3? The method can achieve a comparable result even without ego-state information? Can you elaborate more about the result for ours with ego-state information.

---

> ### Author Response · Authors · 2024-11-20
> **Official Comment by Authors**
>
> We thank Reviewer z3jn for their insightful and positive review.
>
> > The improvement with and with-out pre-training seems not very significant. The module sometimes hurt performance (In tab.1).
>
> Thank you for pointing this out. We want to address your concerns in two aspects:
>
> * Regarding **the performance declines**, we elucidate the underlying reasons by employing a more reasonable evaluation metric.
> * Regarding **the not-so-significant gains from pre-training**, we provide some explanations and conduct further experiments on a larger dataset to better showcase the advantages of our approach.
>
> The results presented in our table demonstrate the **overall enhancement** of prediction performance brought about by pre-training, benefiting both **small objects** (such as *motorcycles* and *pedestrians*) and **large dynamic objects** (such as *buses* and *construction vehicles*). Conversely, **performance declines** are mainly concentrated within **large static categories** (such as *driveable surface* and *manmade*).
>
> We believe that the underlying reason for this situation is rooted in common practices during evaluation, which involve using visible masks to calculate mIoU only for voxels within the visible region. As a result, existing models tend to predict thicker surfaces for large static categories. **Since the current metric does not penalize predictions outside the visible region, models often achieve higher scores through this strategy.** However, this can result in the generation of unreasonable scene structures in predictions.
>
> To address this issue, SparseOcc [1] introduces **the RayIoU metric**, which involves emulating LiDAR rays and projecting them onto 3D occupancy predictions and ground truth to obtain semantic labels for intersected voxels and the traveled distance of the rays as depth information. The semantic labels are utilized to calculate the IoU between predictions and ground truth for each semantic class, with multiple thresholds (1m, 2m, 4m) set to constrain the difference in depth between predictions and ground truth.
>
> **The benefit of using RayIoU is that the model can no longer gain an advantage by predicting a thicker surface.** Therefore, we believe it is a more reasonable evaluation metric for large static categories. We report the detailed performance on large static categories in the table below:
>
> | Method         | GT    | RayIoU   | barrier  | driveable surface | other flat | sidewalk | terrain  | manmade  | vegetation |
> | -------------- | ----- | -------- | -------- | ----------------- | ---------- | -------- | -------- | -------- | ---------- |
> | RayIoU_1m      |       |          |          |                   |            |          |          |          |            |
> | PreWorld       | 3D    | 30.0     | 39.4     | 56.4              | 24.2       | 25.5     | 23.8     | 32.9     | 23.2       |
> | + Pre-training | 2D+3D | **32.5** | **40.1** | **57.8**          | **29.8**   | **27.2** | **27.3** | **35.1** | **25.9**   |
> | RayIoU_2m      |       |          |          |                   |            |          |          |          |            |
> | PreWorld       | 3D    | 37.2     | 44.6     | 64.4              | 29.2       | 30.7     | 31.3     | 43.4     | 36.0       |
> | + Pre-training | 2D+3D | **39.6** | **45.4** | **65.9**          | **34.3**   | **32.7** | **34.6** | **44.7** | **37.8**   |
> | RayIoU_4m      |       |          |          |                   |            |          |          |          |            |
> | PreWorld       | 3D    | 41.9     | 46.6     | 72.4              | 33.2       | 35.6     | 38.1     | 49.7     | **47.3**   |
> | + Pre-training | 2D+3D | **44.0** | **47.4** | **74.3**          | **38.3**   | **37.7** | **41.1** | **50.5** | 47.2       |
>
> As shown in the table, **the pre-trained model surpasses the model without pre-training on almost all large static categories across all thresholds.** The results under the RayIoU metric indicate that pre-training steers the model towards predicting more plausible scene structures. Therefore, overall, we believe that pre-training has not compromised model performance but has yielded gains across various aspects. In summary,
>
> * For **small objects** and **large dynamic objects**, pre-training can lead to significant performance boost.
> * For **large static categories**, pre-training can encourage model to generate more reasonable predictions, which can be reflected in the RayIoU metric.

---

> > ### Author Response · Authors · 2024-11-20
> >
> > As for the not-so-significant gains from pre-training, on one hand, we attribute this to the reasons mentioned earlier, where **the current evaluation metric may not accurately reflect performance on certain categories**. When employing RayIoU as the evaluation metric, as illustrated in Table 7 in the supplementary material, pre-training resultes in a gain of 2.3 RayIoU (36.4 -> 38.7), underscoring the effectiveness and robustness of our approach.
> >
> > On the other hand, we believe this is partly due to **the limited scale of the nuScenes dataset**, which hinders the demonstration of the advantages of pre-training with large-scale 2D data. As a result, we conduct more extensive experiments on the nuPlan dataset, utilizing 3D occupancy annotations from the OpenScene benchmark. Specifically, we randomly select 750 and 150 scenes for training and evaluation. For pre-training, we randomly select supplemental scenes to increase the total pre-training data scale to 1,500 and 1,5000 scenes. **It is noteworthy that we only employ the images from these scenes.** The results are shown as follows:
> >
> > | Method                            | IoU               | mIoU              |
> > | --------------------------------- | ----------------- | ----------------- |
> > | OccNet-T [2]                      | 32.82             | 11.52             |
> > | OccNet-B [2]                      | 34.22             | 13.60             |
> > | **PreWorld (Ours)**               | 38.25             | 18.63             |
> > | **+ Pre-training (750 scenes)**   | 38.40 (+0.15)     | 19.06 (+0.43)     |
> > | **+ Pre-training (1500 scenes)**  | 38.55 (+0.30)     | 19.29 (+0.66)     |
> > | **+ Pre-training (15000 scenes)** | **38.62** (+0.37) | **19.85** (+1.22) |
> >
> > These results demonstrate that as the scale of pre-training data increases further, our approach can lead to more significant performance improvements, thus providing stronger evidence for the effectiveness of pre-training.
> >
> > [1] Liu, et al. "Fully sparse 3d panoptic occupancy prediction." ECCV 2024.
> >
> > [2] Tong, et al. "Scene as occupancy." ICCV 2023.

---

> > > ### Author Response · Authors · 2024-11-20
> > >
> > > > The ego state-conditioned module lacks of sufficient analysis and comparison.
> > >
> > > Thank you for bringing that to our attention. In Section 3.1 and Section 3.2, we compare our proposed forecasting module with the modules utilized in existing 3D occupancy world models. Current methods **separate occupancy prediction and forecasting into two distinct procedures**: the former relies on a frozen occupancy network, while the latter depends on a complex module involving extensive encoding and decoding steps to **obtain future occupancy from current occupancy**. In contrast, we consider **unifying the procedures of occupancy prediction and forecasting for joint training** to be a more natural and end-to-end approach. Therefore, our choose to **obtain future volume features from current volume features**, which is achieved with just a few simple MLPs without the need for excessive design. The results in Table 2 demonstrate that our simple design has already surpassed existing 3D occupancy world models, validating the soundness of our approach.
> > >
> > > In Table 6, we conduct an analysis of the module's impact. The comparison between Row 1 and Row 2 demonstrates that **this straightforward design can still introduce future forecasting capabilities to our models**. Rows 3 and 4, on the other hand, show that this capability can be further enhanced by incorporating ego-state information and leveraging self-supervised pre-training.
> > >
> > > In general, **we introduce this module to offer a more rational framework for 3D occupancy world models**, and the experimental results indicate that we achieve promising outcomes without employing complex designs. We will provide a more detailed analysis of our current experimental results to better articulate this idea, and thank you for pointing this out.
> > >
> > > > "Our Copy&Paste" in Tab.6 is confusing for me. What is the setting of row 2 in Tab. 6 and what is difference between row 2 and copy & paste?
> > >
> > > "Our Copy&Paste" refers to the process of taking our best prediction result on the 3D occupancy prediction task (as shown in the last row of Table 1) and calculating the mIoU between these results and the ground truth of the future 1s, 2s, and 3s. This can serve as a lower bound to showcase the performance of a model without any future forecasting capability on the 4D occupancy forecasting task.
> > >
> > > The setting of Row 2 is that we utilize our forecasting module without introducing ego-state information and self-supervised pre-training. The performance difference between Row 2 and Copy&Paste demonstrates that simply by introducing a strightforward forecasting module and applying temporal 3D occupancy supervision to the model, we can equip the model with future forecasting capabilities and yield non-trivial results (with an average mIoU of 8.25 over 7.79).
> > >
> > > > How to correctly interrept the result in Tab.3? The method can achieve a comparable result even without ego-state information? Can you elaborate more about the result for ours with ego-state information.
> > >
> > > When we do not incorporate ego-state information as input , the model relies **solely on historical image inputs** to predict the ego trajectory. Introducing ego-state information provides the model with additional scene information, such as **velocity and acceleration**, enhancing the motion planning capabilities further.
> > >
> > > For fair comparison, we provide the results of these two settings. The results without ego-state information illustrate that our model can achieve comparable L2 performance to some planning models, albeit slightly higher collision rates. By incorporating ego-state information as input, the motion planning capabilities of our model are significantly enhanced, surpassing other existing 3D occupancy world models following the same configuration by a notable margin.

---

> > > > ### Author Response · Authors · 2024-12-02
> > > >
> > > > We believe we have sufficiently addressed your questions and concerns. With only two days remaining for discussion, we are eager to receive your response. Thank you!

---

### Author Response · Authors · 2024-11-20
**Thank you for your time and insightful feedback.**

We would like to express our gratitude to the reviewers for acknowledging the strengths of our work, including its intuitiveness (**Reviewer z3jn**, **Reviewer BYPK**), clear writing (**Reviewer z3jn**, **Reviewer fN92**, **Reviewer BYPK**), extensive experiments (**Reviewer z3jn**, **Reviewer GcMx**, **Reviewer fN92**, **Reviewer BYPK**), and superior performance (**Reviewer z3jn**, **Reviewer GcMx**).

In response to the reviewers' questions, we have provided corresponding explanations and conducted further experiments. We are looking forward to further discussions!

---

### Author Response · Authors · 2024-11-26

We would like to thank the reviewers for their insightful feedback, which we believe helped to significantly improve the manuscript.

We have included the responses to each reviewer’s comments and highlighted the changes in our revised manuscript in blue. Again, we really appreciate the constructive remarks and thank you all for your detailed feedback.

---

### Meta-Review · Area_Chair_UnPm · 2024-12-20

**Metareview:**

This paper introduces PreWorld, a semi-supervised, vision-centric 3D occupancy model for autonomous driving. It employs a two-stage training framework that integrates 2D labels during a self-supervised pretraining phase and 3D occupancy data during a supervised fine-tuning phase. The work aims to improve 3D occupancy prediction and 4D forecasting while maintaining scalability and leveraging cost-effective data annotation. In summary, reviewers believe this paper exhibits a novel training paradigm and complete experiments. Weaknesses include the incremental contributions, some initial concerns on the evaluation, and paper presentation. However, after the rebuttal, reviewers are all positive about this paper and concerns are (mostly) addressed. Therefore, I recommend accepting this paper.

**Additional Comments On Reviewer Discussion:**

Reviewers initially raised several concerns including: incremental performance improvement (z3jn, GcMx), ablation on different modules (BYPK), model complexity (BYPK), paper presentation and missing details (almost by all reviewers).

Authors provide additional experiments to address these concerns, which are confirmed by reviewers. Paper presentation has been improved in the revision and authors promise to incorporate all comments from reviewers into the final draft.

---

### Decision · Program_Chairs · 2025-01-22

Accept (Poster)